# Dendritic Polymers as Promising Additives for the Manufacturing of Hybrid Organoceramic Nanocomposites with Ameliorated Properties Suitable for an Extensive Diversity of Applications

**DOI:** 10.3390/nano11010019

**Published:** 2020-12-24

**Authors:** Marilina Douloudi, Eleni Nikoli, Theodora Katsika, Michalis Vardavoulias, Michael Arkas

**Affiliations:** 1Institute of Nanoscience Nanotechnology, NCSR “Demokritos”, Patriarchou Gregoriou Street, 15310 Athens, Greece; h.nikoli@inn.demokritos.gr (E.N.); theodorakatsika@gmail.com (T.K.); 2PYROGENESIS S.A., Technological Park 1, Athinon Avenue, 19500 Attica, Greece; mvardavoulias@pyrogenesis-sa.gr

**Keywords:** dendritic polymers, ceramic compounds, biomimetic, nanomaterials, dendrimers, hyperbranched polymers, silica composites, hybrid materials

## Abstract

As the field of nanoscience is rapidly evolving, interest in novel, upgraded nanomaterials with combinatory features is also inevitably increasing. Hybrid composites, offer simple, budget-conscious and environmental-friendly solutions that can cater multiple needs at the same time and be applicable in many nanotechnology-related and interdisciplinary studies. The physicochemical idiocrasies of dendritic polymers have inspired their implementation as sorbents, active ingredient carriers and templates for complex composites. Ceramics are distinguished for their mechanical superiority and absorption potential that render them ideal substrates for separation and catalysis technologies. The integration of dendritic compounds to these inorganic hosts can be achieved through chemical attachment of the organic moiety onto functionalized surfaces, impregnation and absorption inside the pores, conventional sol-gel reactions or via biomimetic mediation of dendritic matrices, inducing the formation of usually spherical hybrid nanoparticles. Alternatively, dendritic polymers can propagate from ceramic scaffolds. All these variants are covered in detail. Optimization techniques as well as established and prospected applications are also presented.

## 1. Introduction

With the advent of nanotechnology and the introduction of nanoscale matter, revolutionary progress was observed in the field of materials science. The synthesis of nanoscale analogues to conventional implements with enhanced properties and a broad spectrum of applications became a routine process. Nanocomposites confirmed the hypothesis that size and shape play a pivotal role in the determination of their exquisite physical and chemical profile. The establishment of superiority over conventional microcomposites was an expected consequence. The exceptionality of these materials originates from their multilayer, core-shell-surface architecture. Depending on their core substance, they can be classified into three key categories: ceramic nanocomposites, metal nanocomposites and polymeric nanocomposites [1,2,3].

Ceramic materials are considered as an emerging trend of recent decades due to their mixed and flexible features, their porous, small-size structure and their broad spectrum of applications. They can appear in various forms, with the most prominent ones being ceramic membranes, ceramic nanostructures and biogenic ceramics, each form possessing exceptional uses and serving different purposes. Their capability to adopt controlled and diverse combined conformations through easily accessible, low-cost, ambient temperature preparation methods render them excellent candidates for the production of hybrids with optimized properties such as core-shell nanoparticles. They also constitute excellent alternatives in order to avoid the undesirable defects of their elementary equivalents, for instance toxicity, instability and high solubility [4,5,6,7].

Among the numerous properties of ceramic membranes, the most distinguished are mechanical tolerance, chemical inertia, thermal stability, damage impedance, energy conservation, high hydrophobicity, satisfactory flux at low pressures and an excessive working lifetime [6,7,8]. All these characteristics contribute to the suitability and expanded applicability of these membranes in various fields such as gas separation [9,10,11] and filtration technology [7,8], water decontamination processes, water and wastewater treatment, desalination [12,13,14,15,16] and catalysis [11,17]. Another advantage of ceramic membranes is their employment in both microfiltration (MF) and ultrafiltration (UF) processes and less frequently in nanofiltration (NF) and reverse-osmosis (RO) [18]. Specifically, for water and wastewater purification, depending on the targeted configuration (flat-sheet or tubular) and the overall properties of the fabricated membrane, the preferred ceramic materials used are alumina, titania, zirconia and silica. Still, the non-specialized nature resulting primarily from their inability to selectively extract ultra-low-sized, “mobile” contaminants and to achieve accurately desired pore size distribution, remains one of their major drawbacks [6,19].

Common problems of unprocessed ceramics such as brittle fracture, poor toughness and strength degeneration highlight the need to introduce secondary components to these materials in order to reach nanoscale ceramic products with upgraded properties. Alumina, titania, zirconia, silica and hydroxyapatite are again useful candidates for the construction of ceramic nanoparticles [20]. These nanomaterials are unique in terms of biocompatibility. Thanks to their surface modification, encapsulation and interaction with biological molecules, such as proteins, lipids and antibodies, they serve as novel diagnostic and therapeutic tools. Some examples include drug-delivery vehicles, biomedical imaging and photodynamic and photothermal therapy [21,22,23,24,25]. In addition, advantages linked to biocompatibility, such as increased dissolution [26], resistance to microbes, stable porosity and high selectivity, along with the facility of synthesis procedures without requirement of toxic substances and extreme conditions, make nano-ceramics the epicenter of pharmaceutical research. They combine safety with the ease of administration and controlled drug availability [20,27,28]. For example, the photooxidative and cytotoxic features of titania can be efficiently manipulated for the fabrication of thermally stable anatase nanoparticles. Therefore, they can be employed as excipients and in the context of photodynamic therapy for the selective treatment of several types of tumors. On the other hand, silica mesoporous nanoparticles can be readily functionalized with proteins and radioisotopes (e.g., radioiodine) to facilitate cancer detection and subsequent radiotherapy [2,3]. Similar forms can also be combined with titania analogues and yield core-shell nanomaterials with advanced catalytic and absorbing performance, as well as anti-pollutant properties [29].

Another category, bioceramics, is considered a convenient type of biomaterial in terms of applicability in biomedical engineering fields. Their biological importance stems from their surface similarities with living tissues and molecules with osseous origin, facilitating therefore their interaction with bones and teeth and thus their involvement in orthopedic therapy and tissue engineering [30,31,32,33,34]. To date, modern bioceramic products replace first-generation inert predecessors claiming the overthrow of conventional tissue damage treatment regimes, since they provide novel therapeutic approaches that rely on tissue repair and replacement in parallel and less on simple substitution. Second generation bioceramics include varieties of different crystallization degrees. They are characterized by high bioactivity, biodegradability and resorbability and are ideal for repairing purposes where a strong and controlled bond with the implant is required. These materials may be based on apatite sources (hydroxyapatite, calcium phosphate mixtures, bone cements based on calcium salts) that function as bone imitators against bone loss or as surface coatings for better bone deposition rates. Alternatively, they may contain a glass source (bioglasses, glass-ceramics) that stimulate bone growth processes by interacting with body fluids or by filling bone cavities. Likewise, third generation counterparts introduce a porous framework for the construction of scaffolds with an adjusted size and shape to accommodate specific molecular components [31,34]. Silica-based mesoporous structures and organic/inorganic hybrid composites intervene in differentiation pathways of mesenchymal cells and regulate osteogenesis and thus promote bone regeneration by exposing active molecules, such as growth factors, hormones and cells at the site of damage [30,31,34].

There is a constantly increasing interest towards the construction of new embodiments that are inspired by living systems, mimicking natural molecules and recapitulating key cellular and molecular processes that have prompted nanotechnologists to incorporate molecules bearing a dendroid architecture into their constructs. Dendritic polymers are ubiquitous materials, graced by exceptional physicochemical properties, well-defined highly branched discoid or spherical shapes and an adjustable surface area. They constitute the fourth major class of polymeric architecture and are classified as (a) monodispersed symmetric dendrimers named after the Greek word δένδρο for tree because of their structural resemblance; (b) asymmetric species, called hyperbranched polymers; (c) fragments of both categories defined as dendrons; and (d) dendrigrafts, which are combinations of conventional (usually linear) polymers with dendritic patterns [35,36]. Their uniqueness emanates from their separate structural units (Figure 1): the core which is the substrate of the radial functionalization. The interior, where the “branches” reside, create a well-suited environment for the encapsulation of guest elements. The exterior part, forming a peripheral surface, functions as an active nano-scaffold and interacts with the nearby functional groups and environmental stimuli. Some of the most established dendrimers are diaminobutane poly(propylene imine) DAB/PPI and poly(amidoamine) PAMAM (Figure 2), while their common non-symmetric hyperbranched relatives include polyglycerol PG and poly(ethylene imine) PEI (Figure 3) [37,38,39,40]. Dendritic polymer’s biomimetic behavior favors their applicability in clinical fields [41,42] as drug carriers, targeted and controlled delivery vehicles [43,44,45,46,47] for anticancer therapeutics [48], in photodynamic therapy [49], imaging [50], diagnostics [51,52] as sensors, gene transfer agents or as artificial molecules per se [30,35]. Additional applications are encountered in the fields of separation systems [36,37], chromatography, membranes, hydrogels [38,39,40,41,42,43,44], solvent extraction [45] and color chemistry such as dyes [46].

Undoubtedly, the inauguration of interventions in the nano-scale gave the opportunity to access new conformations, synthetic or semi-synthetic, of augmented functionality, tractable structure and enhanced multi-functionality, compared to their raw counterparts. Still, the margins for improvement of nanosynthetic constructs will remain large, as long as there are unexplored aspects of different combinations of distinctive precursors. The fabrication of novel, tailor-made hybrid products profiting from combinatorial properties of suitably functionalized dendritic polymers and ceramic compounds represent powerful tool of the utmost importance. The resulting blends have the potential to substitute their unilateral sources, by considerably eliminating most of their inherent defects and benefiting from their individual assets. An inorganic/organic (ceramic/dendrimer) combination for instance can incorporate stabilizing and guest-host characteristics of the polymer matrices, in parallel with the mechanical superiority provided by the ceramic substrate [53]. The yielding organoceramics possess a confirmed potential for a multiplicity of applications, among them the most outstanding include catalysis [29,54,55], water treatment [56,57,58,59] and magnetoceramics [60]. An overview of the prevailing methods used for the synthesis of organoceramic materials will be presented in the framework of this review, along with a summary of potential barriers encountered and the proposed solutions for each given case. The most important applications of these materials will be also thoroughly outlined with a final note concerning the outlooks for the future.

## 2. Methods of Preparation

### 2.1. Simple Coating by Absorption of Dendritic Additives into the Pores of the Ceramic

One of the easiest and fastest methods to obtain hybrids that follow a dendritic/ceramic pattern, is the direct immersion of the inorganic support possessing suitable porosity into a solution of the selected polymer [61]. In this manner, complex organic/inorganic formulations are attained that incorporate the functionalities of both constituents: (a) elevated and selective encapsulation properties, due to the highly accessible empty cavities formed by the branches of the dendritic compound; (b) perfect mechanical properties and absorption/filtering capacity that ceramic supports of variant porosity sizes and shapes can furnish. The most astonishing characteristic of this specific technique is the high polymer impregnation. Loading efficacy depends mostly on the structural components and the resting free surface area of the ceramic substrate, as well as the functionalized polymer compatibility, in terms of reactivity, with the given porous ceramic.

A first typical immersion of a silica monolith to a PAMAM G4 solution yielded a solid anion exchange polyelectrolyte gel for Prussian Blue (PB), iron(III) hexacyanoferrate(II), and cobalt(II) hexacyanoferrate(II) [62]. G2, G4 and G6 PAMAM with silica substrates have been employed in order to take advantage of the electrostatic attractions of positive charged amino groups and mount a second layer of DNA molecules for gene therapy applications [63]. Alternate layering of PAMAM and chondroitin sulphate onto mesoporous silica nanoparticles of an intensified negative charge due to the presence of a primary substrate of carboxylated PAMAM G1 led to ideal carriers for protracted release of doxorubicin and curcumin [64].

After initial capping of the silanol groups of the external surface by (CH_3_)_2_SiCl_2_, G1-G3 PPI dendrimers peripherally modified by amidoferrocenes were incorporated by intermolecular hydrogen bonds to the pores of mesoporous MCM-41 silica affording hybrid redox-active ceramics [65]. Another distinguishing example that employs this impregnation approach concerns alumina filters of distinctive porosity directly immersed into an alkylated dendritic polymer solution under mild heating conditions. Octyl functionalized poly(ethyleneimine) (PEI), the proposed hyperbranched polymeric additive, serves as a cheap and facile-to-synthesize alternative to symmetric dendrimer “nanosponges”, exhibiting equivalently excellent pollutant inclusion constants (*k_inclusion_*) [66]. It creates a highly homogenous water-insoluble coating, a thoroughly efficient depollution film. The hallmark of this technique is the attainment of a polymer impregnation percentage of up to 22%. Interestingly, there is an analogy between the concentration of the dendritic polymer and its impregnation percentage into the same alumina filter, meaning that an increasing concentration (0.2–30% *w*/*w*) of the alkylated PEI allows better impregnation percentages until a certain critical point. When this threshold is surpassed, small pores clog up and no additional polymer can be integrated. This critical point that marks the loading capacity of the specific ceramic compound depends primarily on the compositional characteristic of its pores. The efficient coverage of the ceramic pores is confirmed by the linearity of the plots correlating with the impregnation percentage/concentration ratio as a function of the surface area of the given alumina tube filter.

Another instance that adopts the elementary impregnation approach involves the fabrication of titania processed by PAMAM dendrimers adsorbed into its surface pores through a slurry mixture preparation [67]. The yielding nanocomposites benefit from the non-toxicity of the organic layer and the metal chelation properties introduced by functionalization of the periphery with hydroxy-terminal groups. Herein, only a partial, localized coating of the pores is achieved, driven by the electrostatic attractions developed between the hydroxyl groups of the dendrimer and the surface of titania (Figure 4). These intermolecular forces favor the immobilization process during which the dendritic molecules occupy a rather limited space compared to the total porous volume of pure titania. Specifically, this method can reach an impregnation level of 1% organic content by weight to the inorganic substrate. The composites withstood 2 h at 300 °C, exhibiting minimal mass loss of their active layer (0.16%), proving their high thermal stability.

As expected, the impregnation rate does not depend only on the polymer concentration. The chemical nature of the selected dendrimer is a crucial factor too. The molecules are attracted to the inorganic surfaces due to intermolecular, particularly electrostatic, forces [68]. Typically, accumulation increases with increasing dendrimer generation while it is considerably larger for those with negatively charged groups (e.g., carboxylic anions) on positively charged surfaces (alumina) than that of analogues containing positive charged groups (e.g., ammonium cations) on negatively charged surfaces (silica) [69]. For this reason, the pH and the ionic strength of the solution exert an important influence in as much as they affect the charge of the molecules and the ceramic support [70]. Organic loading is also governed by the pore parameters of each particular ceramic [71]. This was verified by a study that evaluated the impregnation degree of silylated dendrimers and hyperbranched polymers into three ceramic filters of different origins and compared them with the homologous β-cyclodextrin derivative [58]. The resulting data demonstrated that ceramic membranes manufactured by Al_2_O_3_, TiO_2_ or SiC, when immersed into a solution of triethoxy silyl dendritic polymers of the same concentration (30%), yielded hybrid organoceramics of different polymer contents (Figure 5). All of them, however, exhibited higher saturation limits compared to the β-cyclodextrin derivative. As was anticipated, ceramic filters with the highest porosity values (Al_2_O_3_ and SiC) scored the best inclusion values, while the ones with the lowest porosity values (TiO_2_) presented considerably inferior impregnation properties. Pore size also regulates polymer incorporation degree considering that smaller pores allow a wider pore surface area. Therefore, the impregnation percentage of ceramics having the same porosity is determined according to the pore size. The smallest pore size, such as in alumina, in that case leads to a higher content in additives. The type of guest molecules also plays a crucial role to the extent of their incorporation, as each specific solution is characterized by different viscosity. This apparently explains the high impregnation percentages obtained by ethoxysilated polyglycerol compared to the other polymers employed in the context of this study.

### 2.2. Attachment of Polymers with Covalent Bonds to the Surface of the Ceramic Substrate

The performance of the nanocomposites is largely dictated by their surface features. A well-defined and fully characterized external surface leads to better prediction and management of the resulting material properties, such as active ingredient solubility/release and reactivity with the other active factors present. Chemical reactions with active groups induced on the ceramic, for instance by etching with strong acid or base are the conventional, versatile method to obtain highly specialized and flexible modified surfaces. Dendritic polymers, known for the high local concentration of exposed external surface groups, branching diversity and hence susceptibility to chemical interventions, represent unique options to afford hybrid materials with desired chemical profiles. The functionalization of dendritic polymers with groups based on silicon, has gained considerable interest. They offer active sites, commonly silanols, which can subsequently react with chemically relative ceramic substrates, forming stable siloxane bridges through curing processes [56,72]. Some of the most prominent advantages of these Si-heteroatom dendrimers over the conventional ones are as follows: (a) unrestrained shaping and tailoring of the dendrimer structure density with Si atoms as the branching centers of the last 2 or 3 generations; (b) efficient diffusion into the ceramic filler or membrane and improved impregnation rate. This results in lower activation energy requirements and secure reduced curing temperatures and cost effectiveness; and (c) a variety of synthetic pathways.

Production of organosilicon-based ceramics can be implemented through methods such as cross-linking, sol-gel processing, pyrolysis and ceramization. During the fabrication procedure, emphasis should be given to the specifications of the selected dendrimer, and more specifically in its molecular weight, which should be as high as possible, in its solubility into the solution of the ceramic precursor and in its stereochemical convenience to perform cross-linking. Chemical stability of the resulting hybrid material depends on the type of formed bonds. Covalent bonds are the most suitable for sustainability. By being stable they provide the option of regeneration without substantial organic material loss and therefore allow repeated use. Formation of chemical bonds between the dendritic polymer and the ceramic substrate can be established through the sol-gel reaction. The effectiveness of this method in producing organoceramics is based on the employment of appropriate functionalized silyl-reactants, particularly alkoxylates, established for their capability to undergo hydrolysis under acidic conditions [59]. Full conversion is crucial for the shaping of the final product, as heterogeneity affects negatively both the compatibility and the structure of the hybrid ceramic. Triethoxylated derivatives of dendritic molecules, for instance DAB, PEI and PG, are susceptible to rapid hydrolysis, delivering silanols, which in turn can react intramolecularly or intermolecularly with adjacent hydroxyl groups via a condensation mechanism (Figure 6a). Dendritic silanols can condensate with the hydroxyl groups of the targeted ceramic surface as well, enabling the formation of a chemically bound film (Figure 6b). Activation of the ceramic surface and formation of hydroxyl groups is a prerequisite for the reaction and can be performed by using several techniques. An interesting activation example proposed by Ottenbrite is the etching through immersion into the 1 M NaOH solution for some hours, followed by extensive washing with methanol and water and drying under vacuum conditions [73]. This process combined with curing at slightly increased temperatures, successfully promoting the development of nanostructured hyperbranched PEI covalently attached to an activated ceramic filter for absorption of mono and polycyclic aromatic hydrocarbons, pesticides, trihalogen alkanes and methyl *tert*-butyl ethers [59]. Implementation of this exact synthetic pathway with G3 PPI dendrimer loaded with ibuprofen or levofloxacin on mesoporous silica nanoparticles afforded antimicrobial agents to be defined as nanoantibiotics [74,75]. Interference of disulfide bridges between sulfur-containing G1–G5 PPI and accordingly modified mesoporous silica nanoparticles loaded with fluorescent dyes (fluorescein disodium salt and carboxyfluorescein) generated redox-responsive stimuli-triggered release carriers [76]. The transformation of disulfides to sulfhydryl groups in the presence of a reductive reagent removes the dendrimers and permits the discharge of the active ingredients.

Antibacterial PAMAM and carbosilane dendrimers and dendrons have been grafted to silica surface too [77], either by mediation of triethoxysilylpropyl succinic anhydride silane (TESPSA) or by direct grafting, biocompatible G2 and G3 carbosilane dendrons bearing ammonium or tertiary amine groups were attached to mesoporous silica nanoparticles in order to serve as nonviral gene delivery systems [78]. Alternative carbosilanes with Si-Cl at the focal point and peripherally modified zirconocene were used as catalysts [79]. Co-condensation of PAMAM dendron (G1.0) containing a triethoxysilyl coupling agent and tetraethoxy silane (TEOS) in different molar ratios with the addition of cetyltrimethylammonium bromide (CTAB) as pore-directing agents, yielded Hg^2+^ absorbing silica-gel sol-gel [80]. Non-activated mesoporous silica required reflux for 12 h in DMF under nitrogen in order to form bonds with triethoxy-sillylated PAPAM dendrons and provides composite nanoparticles for drug delivery (curcumin) and fluorescence imaging [81].

The adjustable external surface and internal cavities of dendrimers renders them extremely selective and prone to interventions aiming to enhance their chemical affinity to specific targets. The application of their derivatives that are prone for sol-gel reactions as sorbents for undesirable substances removal and analytical extractions, such as solid phase microextraction (SPME) and capillary microextraction, is common [82,83]. Deficient immobilization of the polymeric coating on the fused silica capillary is a typical shortcoming encountered in conventional SPME methods, delaying or even blocking full extraction of a wide range of analytes. Sol-gel processing, however, guarantees firm attachment, allowing for improved extraction performance and maximum use of the overall analytical potential. One of the first studies that investigated adsorptive capacity of dendritic molecules in solventless microextraction and polycondensation, introduced benzyl-terminated dendrons as a chemically stable coating on the inner walls of fused silica capillaries. Principles of sol-gel processing were exploited in order to form and maintain a strong chemical connection between the organic coating and the inorganic substrate and achieve the highest possible stability towards thermal and solvent distortions. The attachment of benzyl-terminated dendrons was accomplished by the mediation initially of 3-(triethoxysilyl)ethylamine and then methyltrimethoxysilane. The introduction of ethoxysil groups into the roots of the dendron enables sol-gel hydrolysis and polycondensation, creating thus a dendritic stationary phase bonded to the column walls, characterized by remarkable uniformity and a roughened, porous appearance (Figure 7). Various types of bifunctional poly(phosphorhydrazone) dendrons and dendrimers were likewise modified by triethoxysilyl moieties for grafting to silica and by amines for trapping CO_2_ [84].

Another variation that involves reaction with activated rough surfaces and subsequent sol-gel siloxane bond formation, exploits the selectivity potential of hyperbranched PEI by grafting onto the surface of a magnetic porous ceramic [85]. The selected ceramic powder was composed of Fe_3_O_4_ and SiO_2_ in order to combine and inherit the magnetism οf magnetite with surface adaptability and chemical stability of silica. This powder was efficiently prepared through dispersion polymerization that involves the coupling of γ-chloropropyltrimethoxysilane. The production of SiO_2_@Fe_3_O_4_-PEI nanocomposites follows a simple two-step process. The magnetic porous ceramic substrate is first activated with methanesulfonic acid, promoting the creation of silanol groups. The latter then underwent a reaction with the corresponding analogues of hydrolized γ-chloropropyltrimethoxysilane at 80 °C to yield a chlorine functionalization reaction. The final step includes the synthesis of the anticipating adsorptive composite by reaction of the porous magnetic ceramic compound with 10% PEI solution at 90 °C for 6 h. In this way, magnetic porous adsorbents with an excessive separation performance and a faster adsorption equilibrium are produced (Figure 8). Another variant implementing magnetic silica (SiO_2_-Fe_3_O_4_) and Pyridylphenylene dendrons adopted two approaches with 3-(iodopropyl)trimethoxysilane or (3-aminopropyl)triethoxysilane (Figure 9) in order to get efficient stabilizing hosts of Pd catalysts [86].

While symmetric dendrimers have been intensively studied for their catalytic properties in homogeneous catalysis, limited information is available for their contribution to the field of heterogeneous or the so-called immobilized catalysts on porous media. A very hopeful area that addresses the insufficiently selective and enantioselective behavior of heterogeneous inorganic supports in asymmetric reactions. It also incorporates the prospect of chiral dendrimers on inorganic supports as catalytic alternatives for enantiomeric reactions [87]. Specifically, the study conducted by Chung et al. describes the application of 3-glycidoxypropyl-trimethoxysilane in the development of silica-supported chiral catalysts based on PAMAM dendrimers (Figure 10) for the enantioselective addition of diethylzinc to benzaldehyde. The chemical grafting reaction of the *n*th PAMAM generation dendrimer derivative onto the silica substrate proceeds under the presence of methanol and reflux for 4 h. The resulting composite can then be further treated to achieve the desired peripheral chiral functionality by reaction with (1*R*, 2*R*)-(+)-1-phenylpropylene oxide and the optional presence of a long alkyl chain spacer.

Implementation of the same concept by the aid of chloropropyl trichloro silane and polyallylamine permitted the coupling of G3 PPI to an amorphous silica gel and the posterior immobilization of palladium nanoparticles for selective hydrogenation catalysis [88]. On the other hand, aminopropyl triethoxy silane (APTES) was required in order to bind polyphosphorhydrazone (PPH) dendrimers on thermally activated silica nanoparticles [89]. Further derivatization of the remaining aldehyde groups with amine-functionalized Poly(Ethylene (Glycol) (Figure 11a) afforded hybrid nanocarriers for silver and silver oxide colloids that exhibit antibacterial activity. Carboxy derivatized glutamic acid-based chiral dendrimers with peptide linkages coupled with APTES gave silica chiral stationary phases for use in HPLC [90], whereas the same reagent in conjunction with 1,10-carbonyldiimidazole was used for binding G1, G3, G4, G5 PAMAM. Their subsequent loading with fluorescein isothiocyanate or rhodamine B isothiocyanate led to photoluminescence sensors for cyanide and copper ions, respectively [91]. Dendritic fragments bearing specialized sol-gel bonds forming groups at their focal point demonstrate another beneficial feature. They serve as convenient media to secure high local concentrations of active ingredients by protracting, for instance, the lifetime of hydrophobic species in aqueous solutions. This stabilizing aspect is confirmed in a study where PAMAM dendrons displayed a protecting effect over water-dispersible TiO_2_ nanocomposites. The establishment of a strong chemical Si-O-Ti bond created a screening polar shell to the solubilized metal oxide from the aqueous environment and thus hampered flocculation [92]. The formation of TiO_2_ nanoparticles, and the chemical attachment of PAMAM dendrons (generation 1–3) carrying siloxy focal points and long alkyl chains (hexyl, C6), was realized through hydrolysis of tetraisopropylorthotitanate and subsequent formation of Ti-O-Si bonds. Both reactions proceeded simultaneously in the same pot, mixing various ratios of the titania precursor and the functionalized dendron in 1-propanol solution at 0 °C and then pouring the mixture in water. The final configuration of dendron-protected titania (Figure 11b) was defined by the metal ion/dendron ratio and the generation of the latter, while its size was only generation-dependent. This functionalization of titania nanoparticles enhanced their photocatalytic activity towards 2,4-dichlorophenoxyacetic acid degradation.

Some other reactive groups may be introduced to ceramic surfaces. Such organosilicates are even commercially available. Silica gels bearing isocyanate or maleimide groups were successfully used for binding PAMAM dendrimers and derivatives thereof [93] for Cu^2+^ absorption studies [94], whereas 3-(1-Piperazino)propyl-functionalized gels presented an ideal substrate to immobilize melamine-based dendrons and were implemented for pesticide (atrazine) containment [95].

### 2.3. Direct Growth of Dendritic Polymers on the Surface of Ceramics

Another strategy for the decoration of the ceramic surface by dendritic sequences is the direct propagation of dendrons on their surface through conventional divergent pathways. This solid-phase synthesis technique is a quick and simple alternative for the preparation of hybrid “dendro-ceramers”. Most of the studies employing this direct growth method are based on silica gels. Nevertheless, the main stages formulating the synthesis schemes that ultimately lead to the product of interest remain the same, both for gels and ceramics: (a) surface activation reaction; (b) dendron core binding; and (c) outspread of the branches.

The grafting of PAMAM dendrimers into the silica surface can be easily achieved by performing three successive reaction steps [85,96]. At the outset, the amino core of PAMAM polymers is introduced to the activated silica surface through the interaction between the silanol surface groups and 3-aminopropyltriethoxylsilane (APES). This reaction is carried out at 110 °C in toluene for 48 h. PAMAM sprouts then proliferate by repetitive additions of a branching unit. Evolution of dendrons on silica surface can be accomplished through a series of successive alternate Michael addition and amidation reactions. Michael addition of methyl acrylate (MA) to the terminal amine generates amino propionate esters. Subsequent amidation with ethylene diamine (EDA) results in the first generation dendron. Repetitive Michael additions and amidations afford higher generations [97]. This primary PAMAM coating of silica exhibited high complexation affinity for Zn (Figure 12a) and equally excellent metal ion absorption efficiency for Zn^2+^ [98], Mn^2+^ [99], Ni^2+^ [100], Co^2+^ [98,101], Cd^2+^ [102,103], Fe^3+^ [103,104] (Figure 12b), Pb^2+^ [105] and U(IV) [106]. Employing exactly the same pattern, Alper et al. grew G4 PAMAM dendrons inside the channels of pore-expanded periodic mesoporous silica [107]. Mesoporous silica particles incorporating PAPAM were used as well for generation-dependent size exclusion chromatography columns [108]. Further treatment of PAMAM dendron external groups with salicylaldehyde [109] enables the effective removal of Hg^2+^ from water, whereas processing by methyl isothiocyanate permits absorption of both Hg^2+^ [110], (Figure 13a,b) and Ag^+^ [111] (Figure 13c) and also from ethanol [112].

Addition of n-octadecylisocyanate [97] leads to the introduction of long aliphatic chains to the periphery of the dendrons (Figure 14). In this way, production of amphiphilic silica bearing a hydrophilic dendritic core combined with a hydrophobic surface is obtained and is designed to deal with lipophilic pollutant encapsulation issues.

In another context, usually the third or the fourth generation of the dendron is required prior to the inclusion of the metal-catalyst in order to amplify its catalytic activity as is the case for recyclable gold nanoparticles that promote oxidation of alcohols [113]. In most instances, an additional stage is required: introduction of complexation sites. Specifically, PAMAM segments are phosphonated in order to establish attractive coordination sites for typical metal catalysts such as rhodium and palladium. Peripherally restructured derivatives can then readily complex with chloro(dicarbonyl)rhodium(I) or dichlorobis-(benzonitrile)palladium(ΙΙ). Resulting composites are applied in catalytic hydroformylation of olefins [114,115,116], carbonylation [117,118,119], hydroesterification [120] and Heck reaction of aryl bromides with butyl acrylate and styrene [121]. Another modification of the PAMAM periphery by salicylaldehyde affords a Schiff base. The reaction of the latter with Mn(OAc)_2_ · 4H_2_O yields an Mn(II) complex, immobilizes Mn(II) onto PAMAM–SiO_2_ hybrids and leads to olefin epoxidation catalysts with increased activity (Figure 15a) [122]. Absorption and crosslinking of external amino groups with cellulase stabilizes and enhances enzymolysis efficiency [123], whereas propagation of up to third generation PAMAM dendrons from silica-coated gold nanorods produces trifunctional composites for anticancer photothermal-chemo-gene therapy and co-delivery of anticancer drugs (doxorubicin) and B-cell lymphoma 2-(Bcl-2)-targeted siRNA [124].

Aziridine polymerization is the prevailing method in order to evolve hyperbranched poly(ethyleneimine) dendrons. Pre-attachment of a reactive intermedia on the ceramic is not required [125,126]. Of particular interest is an unconventional, environmentally friendly technique implementing the dual functionality of compressed CO_2_ as reaction medium and catalyst (Figure 15b) [127].

Another compelling example of this method involves the synthetic pathway of various generations, melamine-based dendrimers directly sprouted on the surface of mesoporous silica. Initiation of this synthesis is performed by a functionalization reaction of the silica support with APES and is followed by succeeding reactions with triazinetrichloride and 4-aminomethylpiperidin (Figure 16). Silica derivatives containing the primary amino groups are alternatively incubated with the solutions of the growth-promoting reagents at 4 °C for 24 h [128]. Replacement of 4-aminomethylpiperidin by tris(2-aminoethyl) amine (TREN) evolves dendrons suitable for CO_2_ absorption [129]. In a related study, combined γ-alumina membranes comprised of a mesoporous γ-alumina layer, grown into a macroporous alumina solid support, undergo a specialized activation process. Direct evolution of the dendrons off the alumina surface is once more triggered by the attachment of APES on the support, and is differentiated by the consecutive treatment by diisopropylethylamine and piperazine instead of 4-aminomethylpiperidin. The obtained modified membrane was further functionalized with dodecylamine in order to augment its potential in polarity-based separation applications [130].

There is no reason for the restriction of the dendritic evolution to conventional forms only. Dendrigrafts are a distinctive and rather exotic class, and the building blocks of the internal branches are replaced by polymeric side chains may equally emanate from a suitable focal point. Tsubokawa et al. produced such a type of coating, starting by aminopropyl triethoxy silane root and “postgraft polymerization” of vinyl groups [131,132]. Although entirely symmetrical branching cannot be attained through solid-phase strategies, still, it furnishes a convenient and inexpensive alternative in order to immobilize dendritic functionalities through a single sol-gel reaction onto diverse ceramic frameworks such as silica, alumina, titania, ceria, lanthania, hafnia and zirconia.

### 2.4. Sol-Gel Cross-Linking of Silyl Dendritic Polymers

Silicon-containing dendritic polymers are a promising subcategory, distinguished for the mixed properties mostly originated from the radially multilayered copolymeric morphology and the distinctive characteristics that each constituent layer impose (Figure 17) [133]. Poly(amidoamine-organosilicon) (PAMAMOS) dendrimers present a hydrophilic poly(amidoamine) (PAMAM) molecular core and a hydrophobic organosilicon surface. They provide different structures as building blocks or active sites for various synthetic pathways depending on the chemical reactivity of their corresponding end-groups. Among them, alkoxysilyl-functionalized groups are susceptible to crosslinking chemistry, giving rise to PAMAMOS highly defined three-dimensional networks (Figure 18) [133,134]. PAMAMOS “honeycomb like” compositions are prepared through conventional cross-linking methods, which enable the formation of organosilicon structures with nanoscale dimensions, tunable sizes and shapes and high applicability in the areas of films, sheets and coatings. These sol-gel reactions follow the same two-step synthetic pattern as previously described for the dendritic moieties on the surface of ceramics: hydrolysis of alkoxysilyl (Si-O-R) end-groups to the corresponding silanols (Si-OH) and condensation of resulting silanol intermediates into final siloxane (Si-O-Si) bridges. The rate of the crosslinking reaction is greatly determined by several external factors and reagents, such as additional catalysts and reaction conditions [133]. For example, the optional presence of a catalyst, such as bis(2-ethylhexanoate)tin and variations to curing periods and temperatures of the solvent-free PAMAMOS blocks, lead to a multiplicity of organic-inorganic nanohybrids with completely different properties depending each time on the chosen conditions. Multifunctional crosslinking agents such as tetramethoxysilane or tetraethoxysilane may be employed to increase crosslink density of the dendrimers and silica content. Moreover, controlled thermal degradation of PAMAM molecules allows the formation of porous nanocomposites, where the final pore size is dictated by the diameter of the selected PAMAM core [135].

Hybrid, silica-containing hyperbranched polymers can be also be produced by the aid of PEIs (MW 5000 or 25,000) and 3-(triethoxysilyl)-propylisocyanate as the silica precursor in a *N*,*N*-Dimethylformamide (DMF) or chloroform solution. The crosslinking density remarkably relies on ethoxysilylated polymer end groups reactivity; hydrolysis and concomitant polycondensation rates, that ultimately drive the architecture of the silica-PEI network. Hybrid products can be exploited, either as they are or impregnated/covalently bound to another appropriate ceramic substrate. They also offer the option of pyrolytic dendritic template removal, affording nanoporous silica powders with advanced textural characteristics. Nanomaterials produced by sol/gel processes display similar morphology characteristics, such as surface areas varying between 300 and 500 m^2^·g^−1^ and a pore diameters of around 2 nm [136]. Cohydrolysis and polycondensation is not limited to dendritic polymers containing solely silicon heteroatoms. Phosphorus dendrons of first, second, and third generation bearing ethoxy silane groups at their apex cross-link with tetraethoxy silane (TEOS) leads to silica xerogels with adjustable textures too [137]. Very recently it has been discovered that even “traditional” (i.e., not containing silicon) hyperbranched PEIs form nanocomplexes via hydrogen bonding with orthosilicic acid produced by the acid hydrolysis of TEOS. These nanostructures grow faster at neutral pH (7.5) and substantially slower at a lower pH (pH 5) and transform to hydrogels. When dried they became organoceramic xerogels. Active ingredients may be incorporated into the dendritic cavities at any step of the process. In the framework of a multinational European Project (Novel Marine Biomolecules Against Biofilm Application to Medical Devices: NoMorFilm) this procedure was tested on titanium microporous coating of a stainless-steel model orthopaedical implants with excellent results [138].

### 2.5. Biomimetic Reactions

Evolution has equipped living organisms with advanced self-constructing capabilities in order to secure better survival possibilities. Self-assemblage biogenic mechanisms are introduced for the development of complex organizations. The process where biological molecules activate and control independently a cellular pathway leading to the construction of metal nanoparticles is called biomineralization [139]. Biosilicification, the analogous procedure for biogenic formation for silica-based materials, occurs naturally in many microorganisms too, either terrestrial (higher plants) or aquatic ones (diatoms, sponges). It proceeds rapidly, under mild aqueous conditions, such as ambient temperature, pressure and neutral pH [140]. Biogenic silica occurs in a great diversity of nanopatterned frameworks, with varying sizes, shapes and organizations that are species-specific and malleable in nature, all deriving from the single reaction of silicic acid condensation [141]. This “manipulating siloxane bridge formation” advantage is attributed to the mediation of several biopolymers [142,143]. Specifically, it is established that post-translationally modified polycationic peptides, such as silaffins and R5 peptides, both isolated, from diatom cell walls, precipitate silica nanospheres almost instantly and in a peptide concentration-dependent fashion. There are numerous subsequent studies that successfully used conventional synthetic substitutes of biomolecules to imitate the evolution of biocomposites. For instance, a variety of homopeptides such as poly(arginine) and poly(lysine) [144] and a combination of triethanolamine and cetyltrimethylammonium chloride [145] can serve as biomimetic templates, forming silica nanoparticles with controlled specifications.

Amine-terminated dendrimers of various generations constitute exceptional templates that stand out for their similarity to proteins and tunable design that is primarily dictated by the high concentration of well-localized nitrogen functionalities, which resemble the units found in their biological counterparts and are responsible for silica nanospheres production [142]. For instance, poly(amidoamine) PAMAM and poly(propylene imine) PPI represent popular matrices for the formulation of nanopatterned silica (Figure 19) and alumina (Figure 20) [146]. The silica ratio of the final aggregated nanoproduct depends highly on the number of amino groups of the selected dendrimer. Knecht et al. described biomimetic sol-gel reactions as a powerful method to produce PAMAM/PPI-Si nanospheres. The sol-gel reaction proceeds at room temperature through the addition of metastable silicic acid into the dendrimer solution. Use of a phosphate buffer is required to obtain a pH of 7.5. Organic (dendritic) scaffolds are encapsulated into the gradually forming silica nanospheres and then the hybrids precipitate from a water solution in a way similar to that described before for the case of bioactive peptides. The outcome of the reaction and the production of nanospheres with distinct and specific dimensions hangs on the appropriate selection of dendritic reagents, with proper amine concentration suitable conditions and the presence of inorganic salts [143]. Given the fact that hydroxyl-terminated PAMAM analogues do not cause silica precipitation, it is ascribed exclusively to the presence of amino groups.

Poly(amidoamine), PAMAM (G5.5) anionic starburst dendrimers functionalized with surface carboxylate groups have also been used as templates for the synthesis of hybrid nanocrystalline ZrO_2_, CeO_2_ and Y_2_O_3_ [147]. These ceramic powders were obtained through hydrolysis of the methylester-terminated dendrimers in the presence of stoichiometric NaOH volumes. The selection of the highest possible dendrimer generation is essential for the inhibition of extensive crystallization and the control of the final morphology of the nanoparticles. Thermal treatment at 600 °C for at least 2 h was a necessary step for the shaping of the amorphous nanoproduct into highly ordered spherical nanocrystalline powders. As long as the encapsulated dendritic polymers and their optional active guests in their internal cavities retain their properties, the resulting nanomaterials may be applied in various processes, including environmental remediation and catalysis. Nanospheres obtained through non-symmetrical analogues serve as inexpensive alternatives for both organic and inorganic pollutant sorption and water decontamination. Hyperbranched Poly(ethylene-imines) were proven ideal matrices as they caused silica nanosphere precipitation even in the absence of phosphate buffer, albeit to a noticeably lesser degree. The resulting hybrids removed toxic heavy metal ions (Pb^2+^, Cd^2+^, Hg^2+^, Cr_2_O_7_^2−^) and polycyclic aromatic hydrocarbons (pyrene, phenanthrene) from water [148]. For the attainment of catalytic activity, templated metal nanoparticle nucleation occurs by reduction of trapped ions prior to inorganic shell organization. Composites containing Au and Ag successfully catalyzed 4-nitrophenol reduction oxidation of methylene blue and selective oxidative transformation of benzyl alcohol to benzaldehyde [149].

An evolution of the above procedure proposes the fusion of two different biomimetic processes, biosilicification and biomineralization in one pot synthesis procedure. Both syntheses are regulated by the same hyperbranched polymer in the role of protein substitute [150]. In this way, silica-PEI-silver nanoparticle composites have been assembled, retaining antibacterial activity, catalytic properties and protracted release profile of the respective metal-dendritic polymer complexes. The synthesis of these highly compartmentalized ceramic-organic metal nanocomposites is attained via two successive steps that use the same functional scaffold: PEI Mr = 25,000 (Figure 21). At first, silver ions absorbed into the PEI pockets undergo reduction to metallic silver (core), then silica precipitation occurs from the silicification in the periphery of the dendritic polymer (shell). The bactericide activity of these hybrid nanoparticles is of particular interest. *E. coli, P. aeruginosa* and *S. aureus* assays established that Si-PEI-Ag nanoparticles efficiently prevented their proliferation, consistent with the concept of conservation of the complexes features despite their enclosure into silica capsules.

Bearing in mind the large impact of electrostatic forces between dendrimers and silica precursors, a similar influence in the size and shape of the biomimetically produced silica particles is rationally expected. Ionic strength plays a pivotal role. In the case of PAMAM, for example, for all generations (G1–G6), it has been established that silica corpuscles get bigger with augmenting phosphate concentrations (Figure 22a) [143]. On top of that, at pH 7.5, when the phosphate buffer was replaced with individual alkali metal-chlorine solutions, SEM analysis unveiled a similar linear increase in particle diameter with increasing concentrations up to 100 mM (Figure 22b). Smaller cations up to K^+^ generated typically larger composites, most probably due to preferential interaction with a single silanol group. In contrast, larger cations tend to affect multiple silanols, provoking imperfect surface coverage and limited charge neutralization. These results highlight further the role of neutralizing agents on the growth of silica nanospheres, i.e., securing as minimal as possible electrostatic repulsions and as much as possible free space for unhindered agglomeration.

Dendritic scaffolds are likely to deliver many roles in the fulfillment of the silica condensation reaction. Generally, polypeptides, polyamines and polyamidoamines are partially protonated (NH_3_^+^ groups) at neutral pH, suggesting that an equilibrium is achieved between the opposing charges of the template and the surface of developing silica [151,152,153]. This neutralization results in the creation of an extensive network, with NH_3_^+^, Si–O^−^ interacting groups and solubilizing water molecules. These interactions also drive the incorporation procedure of the matrices within the silica nanospheres. Overall, silica PEI-nanospheres display a negative potential at low pH that reaches the value of nearly −80 mV at pH 11, as indicated by zeta-potential measurements. In addition, the isoelectric point (IEP) after pyrolysis is nearly 2, similar to the IEPs typically encountered in the case of amorphous silica. The presence of the dendritic core in the nanospheres shifts the IEP towards higher pH values. Indeed, the hybrid organic-inorganic nanospheres show an IEP of roughly 8, persistent with the presence of positively charged amino groups. These hybrid nanocomposites present a positive surface potential of nearly +50 mV at pH 2, that decreases to about −70 mV at pH 11.

Besides creating novel ceramic nanostructures, dendritic polymers can serve as crystal modifiers for the biomimetically controlled production of nanoscale biomaterials. The development of these nanoproducts has attracted increasing interest recently, especially in the field of drug delivery systems and biomolecule expression. Aquasomes, for instance, are nanocrystalline ceramics with flexible surfaces that can be appropriately modified with carbohydrates in order to interact with specialized proteins and pharmaceuticals and to transfer them to tissue-specific areas [154,155]. An interesting relevant study attempts to fabricate spherical hydroxyapatite nanocores (Figure 23) by employing carboxylic acid-terminated PAMAM of 3.5 and 4.5 generations as starting materials for the production of hemoglobin-carrying aquasomes [156]. Noteworthily, hydrothermal treatment with generation 5.5 PAMAM drives the production of highly crystalline hydroxyapatite nanorods [157]. In general, dendritic PAMAMs, due to amido groups in their external surface, induce hydroxyapatite nanoparticles with an adjustable size and shape according to their generation through a hydrothermal crystallization. Increasing PAMAM generation number from G1.0 to G4.0 causes a decrease in the particle size from 82 to 38 nm respectively. In parallel, a transition occurs in the resulting shape from rod-like to ellipsoid-like [158]. Analogous effect is observed with increased concentrations of PAMAM G4.0. On the other hand, their polyhydroxy-substituted analogs in most cases formulate elliptical particles with an average grain size of about 20 nm and rarely short nano-rods with an average length of about 30 nm and an average width of about 25 nm [159].

Cationic fourth generation diaminobutane poly(propylene imine) dendrimers (DAB) strongly favor the formation of hydroxyapatite over all the other possible calcium phosphate crystalline phases [160]. The crystallization process may be regulated by introducing different thermal conditions and dendrimer:calcium ratios (Figure 24). The resulting nanoparticles display a characteristic elongated hexagonal rod-like configuration, with dimensions that are largely dictated by the particular reaction conditions. For example, increasing the temperature of hydrothermal processing from 80 to 130 °C leads to a noticeable lengthening, while maintaining room temperature conditions induces a fine nanostructure with distinctive mean length and width values (i.e., 11 and 5 nm). The presence of DAB dendrimers plays a vital role in the homogeneity of the developing hydroxyapatite nanocrystals, as well as in energy conservation and cost reduction of the process. Conventional hydroxyapatite crystal generation in the absence of dendritic mediators requires high temperatures (i.e., 130 °C) and exhibits a highly inhomogeneous morphology with broad particle size distribution.

Hyperbranched PEI bearing long alkyl chains, when mixed with single-chain surfactants such as octadecylamine, haxedecyl trimethylammonium bromide and sodium dodecyl sulfate afford a well-defined dendrimer/HAP biocomposite that mimics principal structural and mechanical properties of bones, such as plastic deformation and high stiffness and strength. The selected surfactant clearly affects not only the size and shape of the resulting hydroxyapatite composite, but also its mechanical idiosyncrasy, with octadecylamine and haxedecyl trimethylammonium bromide yielding composites exhibiting the highest toughening [161].

## 3. Improvement Strategies for Performance Optimization

As discussed above, the performance of filters in applications such as water purification is directly proportional to the impregnation percentage of the dendritic polymer. The latter is governed by the particular pore architecture of the ceramic by the pH and the ionic strength of the immersion solution but mainly by its concentration [87,162,163]. A common emerging problem during the impregnation process is the rapid pore blocking which is aggravated abruptly with viscous dendrimeric derivatives as concentration of the coating solution raises [61], due to the intrinsic spongy labyrinthine configurations of the porous structures. Another issue encountered at denser immersion solutions is the structural breaks/cracks that mark the final composite product, and can undermine the filter absorption performance. These deformations originate from the stress applied during the drying process. Smoother, flawless, ceramic surfaces with optimum absorption properties (Figure 25) can be attained through a simple multiple immersion technique to solutions of gradually increasing concentrations. This method combines both adequate polymer loading with controllable porosity decrease, and avoids pore blocking and all the other disadvantages of immersion in concentrated media.

The involvement of dendritic polymers in the evolution of original nanoscale species with optimized characteristics is raised continuously. Their full biomimetic potential is exploited in parallel. Knecht et al. were the first to introduce pH regulating agents, such as phosphates or HCl, as a means to achieve optimal ionic strength and pH stabilization, as well as to control the final nanosphere dimensions [142]. The idea of double biomimicry gave rise to the synthesis of more complex inorganic/organic/inorganic nanostructures that generally exhibit combinatorial, enhanced properties [150]. These hybrid materials usually carry a dendrimeric matrix stuffed by metal nanoparticles and surrounded by a ceramic shell. Each single biomimetic reaction is successively and independently carried out in order to retain the full properties of the two constituents. It is possible to micromanage the specifications of the final hybrid product, such as the size and shape of both inner metal and outer ceramic particles, the overall composition and the porosity and pore dimensions of the inorganic segment, by adjusting either the biomineralization incubation time or the ratios of the silicification precursor towards the dendritic matrix or the molecular weight of the latter. For example, higher molecular weight of the organic core causes faster formation of metal nanoparticles and optimal catalytic performance of the final product. Thus, biomimetic process optimization can offer unique hybrid solutions with both combinatorial and manageable structural characteristics, evolving nanomaterial synthesis to a whole new level.

Numerous factors can perturb the perfectly symmetrical growth process of the dendritic polymer onto the ceramic surface, affecting the overall properties of the final nanoproduct. For example, side reactions usually co-occur with the propagation pathway of PAMAM on the silica surface, inducing thus major structural diversifications such as dimer and intramolecular cyclization (Figure 26a) [96]. Specifically, a high density of trihydroxy aminopropyl silane molecules bond to the ceramic surface leads to an equally high concentration of precursor- “dendron core” amino groups, which in turn cause serious formational defects. Lower density of dendron-generating amino groups leads to a decrease in steric hindrance, as well as an increase in the in-between distance of secondary amino groups of the dendritic branches, thus preventing the formation of undesired inter-dendron and intra-dendron cross-linked structures in higher generations (Figure 26b). This effect can be successfully avoided through the co-immobilization of 3-(triethoxysilyl) propionitrile on silica, which serves the role of the “spacer”, reduces the “dendron-core” amino group density and thereby improves the whole evolution procedure.

The non-uniform shape and irregular size of the pores are two other factors contributing to the steric/crowding effect. A prominent solution includes the exploitation of silica bearing highly ordered, hierarchical channels (e.g., hexagonally packed) that shows narrow size distribution, such as MCM-41, and SBA-15 [164,165]. These afford dendron development up to fourth or higher generations with negligible structural defects.

## 4. Concluding Remarks—Perspectives

Hybrid dendritic/ceramics materials offer a multitude of perspectives in a variety of interdisciplinary fields replacing traditional, non-functionalized materials that usually suffer from structural inhomogeneity, absorption instability and thus limited applicability [166,167]. The use of dendritic polymers as additives is now widely accepted. In fact, there are several studies substantiating their superiority as stabilizing agents that do not undermine the physicochemical properties of the guest molecules. Some applications covered herein are summarized in Table 1. A plethora of perspectives are envisaged. For instance, PAMAM polymers have been used as stabilizers of semiconductor quantum dots, CdS synthesized at the nanometer-scale [168]. The stabilization effect lies in the effective solvation of these highly insoluble nanoclusters, avoiding precipitation, which occurs in the absence of PAMAM. Dendritic polymers also stabilize colloidal suspensions of gold by in situ reduction of its ions encapsulated into their cavities [169]. Silver nanoparticles may be obtained too through a reduction of cations encapsulated into dendrigrafts, even without a reductant [170]. Optical and structural characteristics of these nanoparticles, such as UV-Vis absorption spectrum, eventual light emission, size and shape are determined by the chemical nature of the dendritic matrix and the host/guest concentrations ratio. Dendritic polymers are furthermore well-known for their ability to deliver an immense variety of colored metal complexes. The composites with ceramics retain this characteristic, along with luminescence properties according to the metal particle conformation. These hybrid sensors can be particularly useful in the detection of explosive chemical warfare agents, as well as other substances, even at the smallest concentrations.

Moreover, these metal organoceramic nanocomposites are noteworthy candidates for catalysis reactions [171,172,173] (Table 2). They are particularly useful when gas phases are involved in reactions such as low molecular hydrocarbon hydrogenation and epoxidation. Ceramics may act not only as solid supports but also local reactant concentration enhancers, mostly through electrostatic attractions. Moreover, when both components exhibit photocatalytic properties, an enhanced performance is achieved. In some cases, the incorporation of the dendritic polymers increases the porosity of the inorganic supports to the point that they approximate the kinetics of homogenous catalysis. The same phenomenon occurs when inorganic nanoparticles are coated by dendrons. Concurrently, they allow easy and rapid retrieval regeneration and reuse.

Besides, due to their antimicrobial attributes against a wide range of bacteria, viruses, fungi parasites and other microorganisms, they may serve as leather or textile disinfection purposes [150]. Decontamination devices may also benefit from the implementation of these ceramic-organic-metal ternary systems. Thanks to their combined adsorption (Table 3) and microbicide capacity, they can be used for water treatment processes. In contrast to other absorbents, they are immune to bacterial infections and, apart from absorbing toxic pollutants, they may provide in parallel a certain level of sterilization.

Another compelling aspect of dendro-ceramics is that they can be employed in nanomedicine (Table 4) as controlled DNA expression vehicles, emulating the modularity model that viruses follow in order to target distinct regions of the cell [174,175]. Silica nanoparticles modified by aminopropyltriethoxysilane (APES) and methyl triethoxysilane (MTES) can generate ternary complexes with DNA and dendrimers. The dense configuration of silica nanoparticles accelerates DNA accumulation into the surface of the cells and subsequent DNA endosomal-lysomal uptake, leading to better cell transfection levels. A biogenic silica system produced by a complex of DNA and a dendrimer bearing an amino group could simplify the formation of the silica nanoparticles and optimize the overall transfection process, as long as this ternary complex remains non-toxic.

Last but not least, drug delivery systems based on hybrid ceramics bearing dendritic polymers is a hot topic as well (Table 5). There is an established potential for controlled protracted and stimuli triggered release as well as targeted therapy [176]. Shortcomings of dendritic polymer carriers are mainly an undesired response by the immune system [177], rapid clearance from blood circulation due to small size [178] and toxicity of cationic high Mw species [179]. They may be addressed by charge neutralization and magnification, resulting in the combination with silica. On the other side, there are biocompatibility issues on silica nanoparticle forms, such as toxicity due to aggregation or hemolytic activity. These are alleviated either by the formation of the composites or by additional coating by a biocompatible polymer [64].

## Figures and Tables

**Figure 1 nanomaterials-11-00019-f001:**
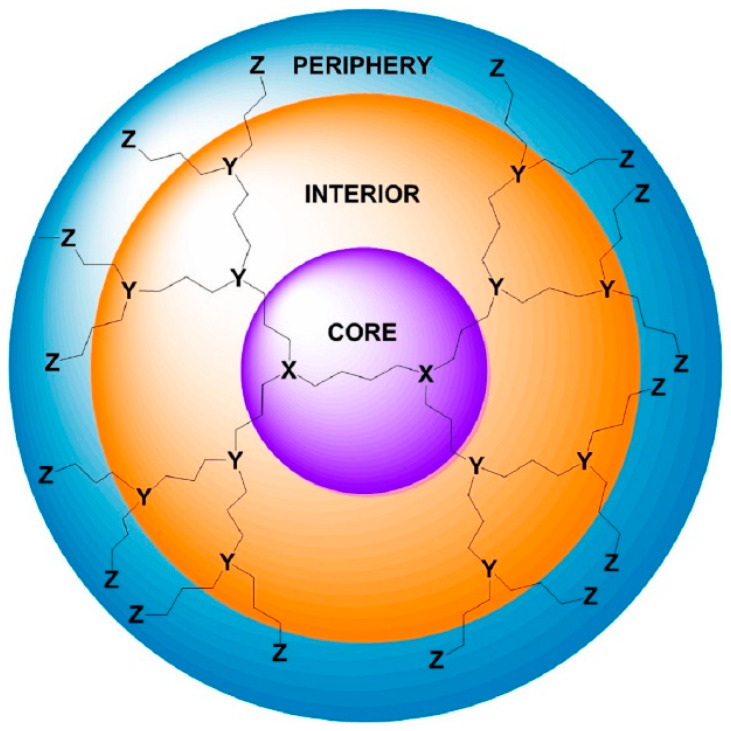
Schematic representation of the three structural parts of a dendritic polymer.

**Figure 2 nanomaterials-11-00019-f002:**
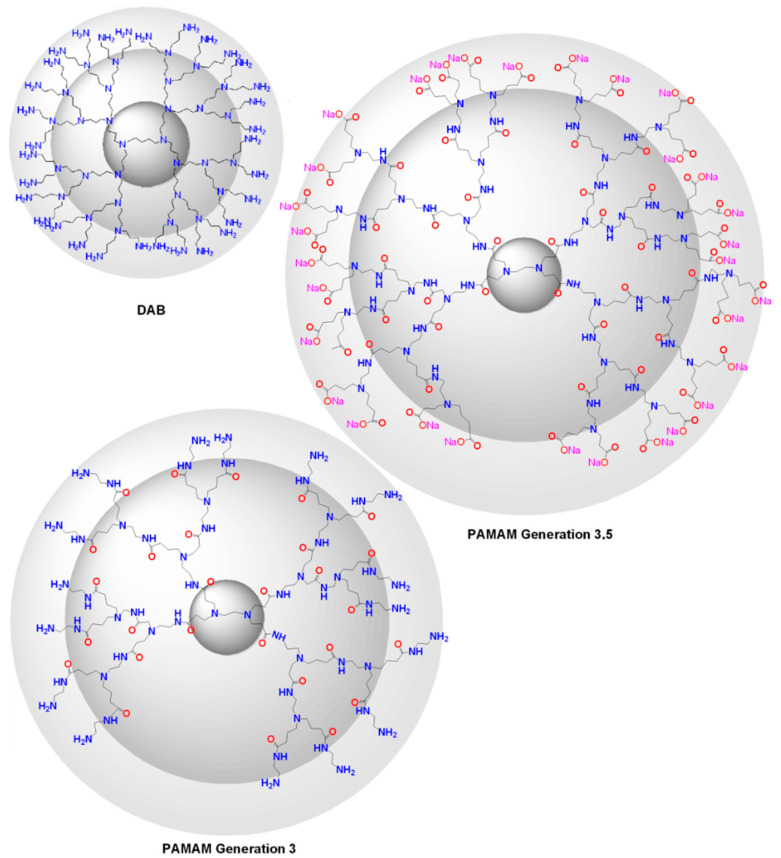
Chemical structures of Diaminobutanepoly(propylene imine) DAB (PPI), and Poly(amido amine) PAMAM, 3rd and 3.5rd generations.

**Figure 3 nanomaterials-11-00019-f003:**
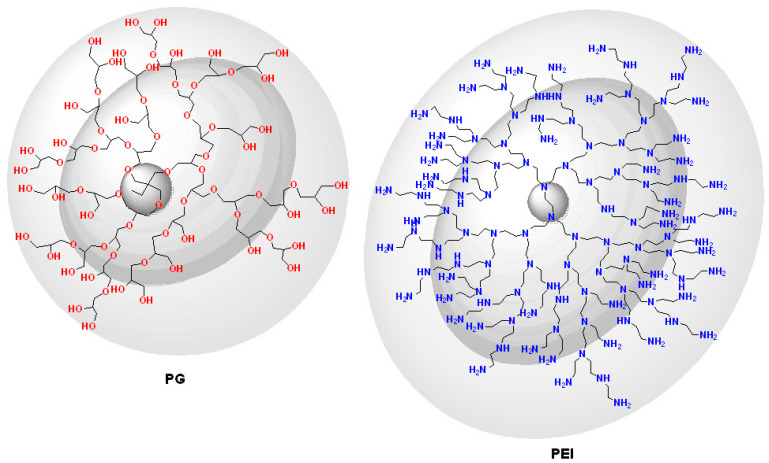
Chemical structures of hyperbranched polyglycerol (PG), and hyperbranched poly(ethylene imine) (PEI).

**Figure 4 nanomaterials-11-00019-f004:**
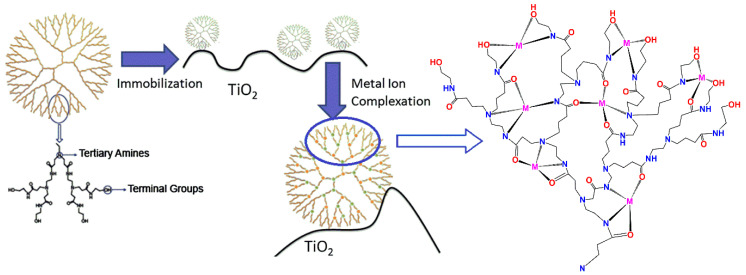
Schematic representation of the immobilization process of G4 dendrimers onto the surface of titania. (Reproduced with permission from [67]; Copyright Elsevier, 2013).

**Figure 5 nanomaterials-11-00019-f005:**
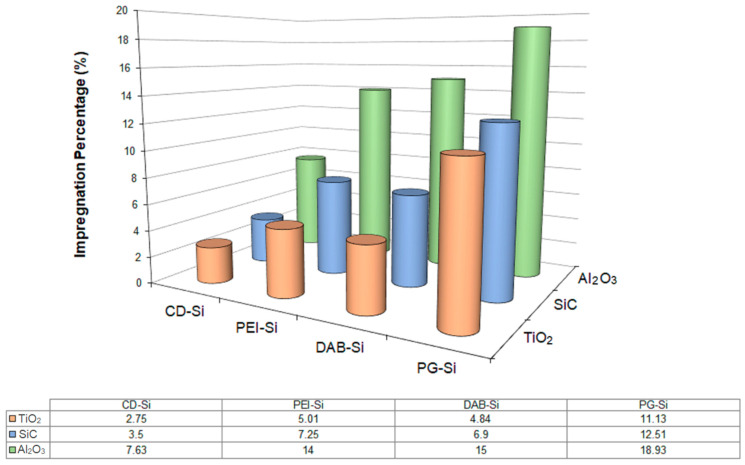
Loading percentage for three ceramic filters of different origins (TiO_2_, SiC, Al_2_O_3_) after their immersion into 30% solutions of triethoxy silated dendritic and cyclodextrin polymers (Reproduced with permission from [59]; Copyright Elsevier, 2006).

**Figure 6 nanomaterials-11-00019-f006:**
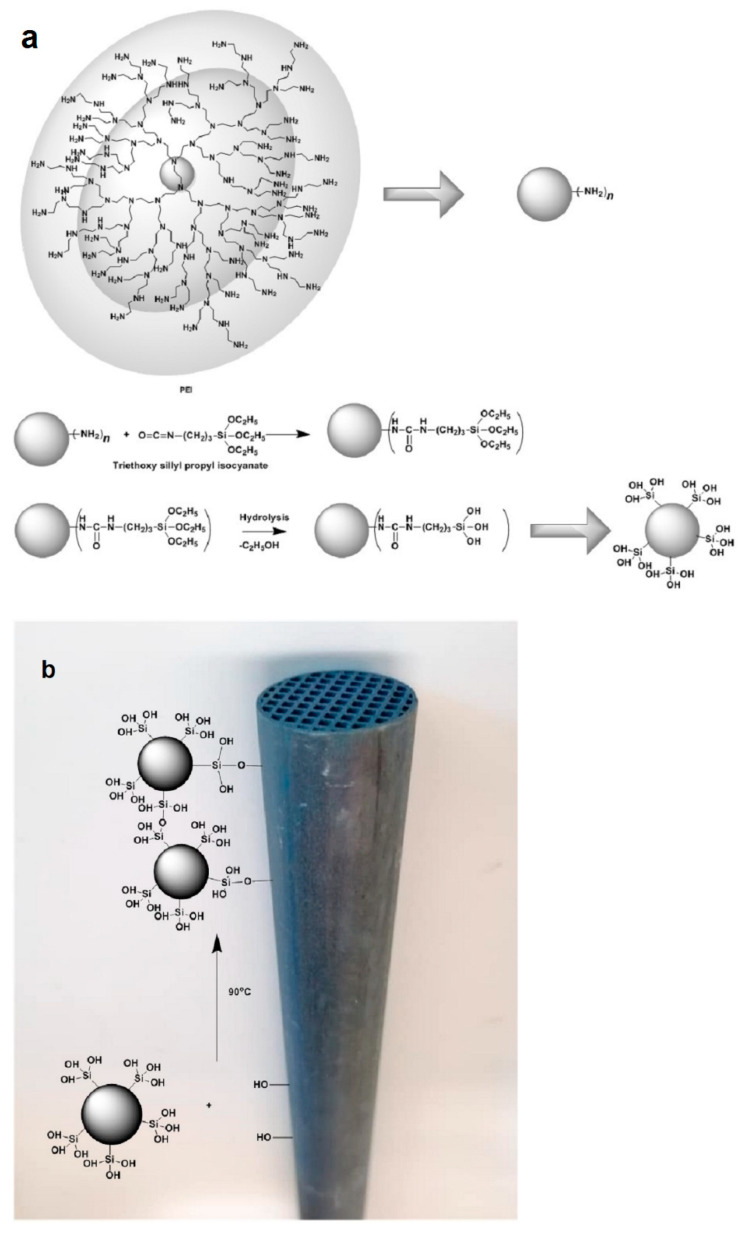
(**a**) Hydrolysis of triethoxysilyl derivatives forming silanols susceptible to polycondensation with neighboring hydroxy groups; (**b**) attachment of functionalized dendritic polymers on the surface of silica ceramic filters by formation of siloxane bridges (Reproduced with permission from [59]; Copyright Elsevier, 2006).

**Figure 7 nanomaterials-11-00019-f007:**
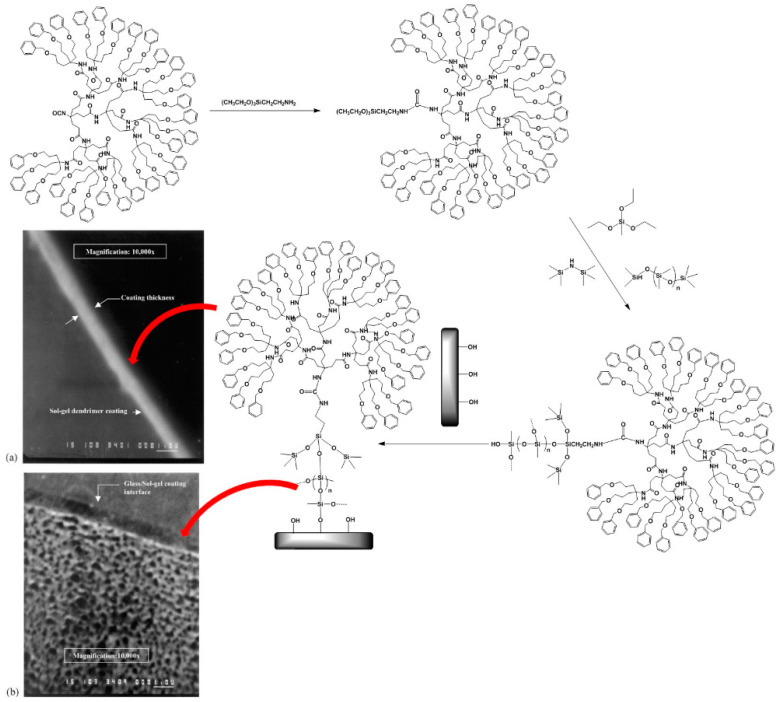
Synthesis of Phenyl-terminated dendrimer with triethoxysilyl root and subsequent immobilization to a fused silica capillary. SEM micrographs of coating thickness (**a**) (0.5 μm) and roughened porous texture (**b**). (Reproduced with permission from [83]; Copyright Elsevier, 2004).

**Figure 8 nanomaterials-11-00019-f008:**
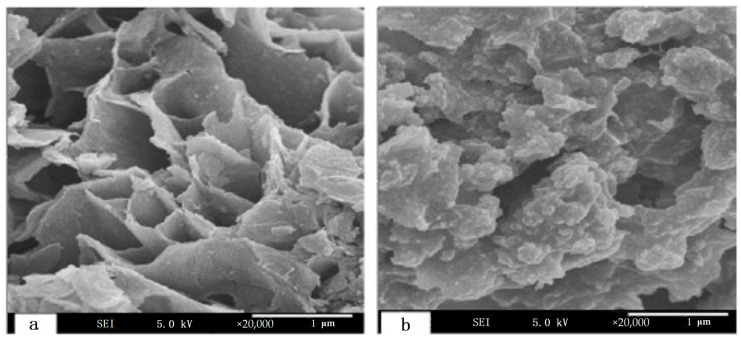
SEM micrographs of Fe_3_O_4_ and SiO_2_ composite magnetic powder (**a**) and PEI-grafted adsorbent (**b**). (Reproduced with permission from [85]; Copyright Elsevier, 2011).

**Figure 9 nanomaterials-11-00019-f009:**
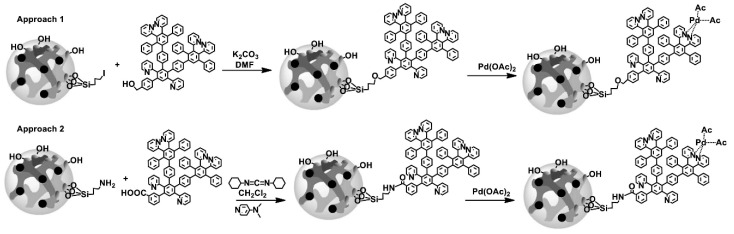
Attachment of Pyridylphenylene dendrons to magnetic silica with the aid of iodine (Approach 1) or amino (Approach 2) groups succeeded by the complexation with Pd acetate. (Reproduced with permission from [86]; Copyright Elsevier, 2019).

**Figure 10 nanomaterials-11-00019-f010:**
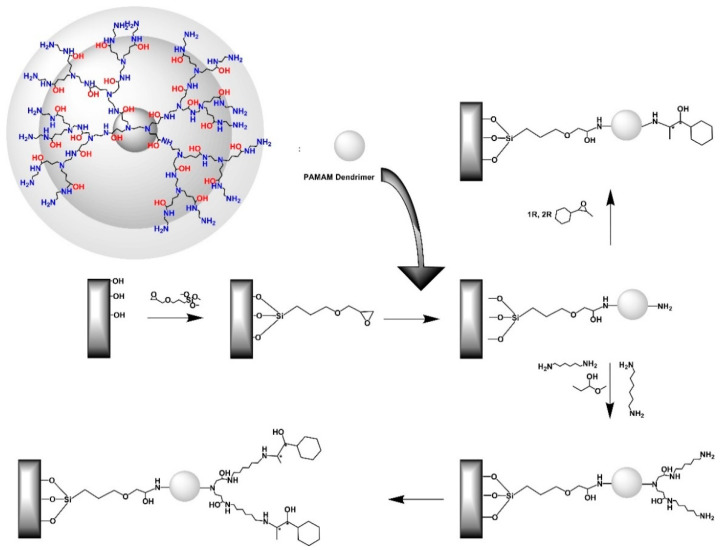
Synthetic path for the immobilization of PAMAM dendrimers decorated by chiral functionality onto silica substrate. (Reproduced with permission from [87]; Copyright Elsevier, 2003).

**Figure 11 nanomaterials-11-00019-f011:**
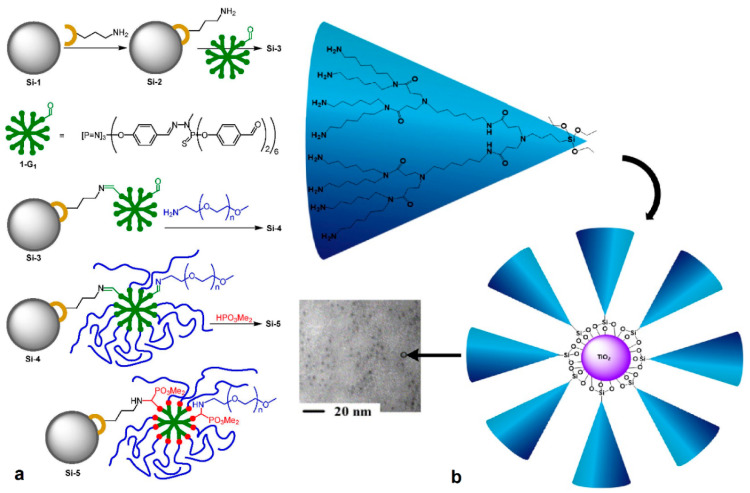
(**a**) Synthetic path for grafting polyphosphorhydrazone dendrimers onto functionalized silica and following modification of its periphery with Poly(EthyleneGlycol) chains. (Reproduced with permission from [89]; Copyright Royal Society of Chemistry, 2013). (**b**) Pathway for the clipping of the PAMAM dendron protective layer to titania photocatalytic nanoparticles and SEM micrograph of the complexes (Reproduced with permission from [92]; Copyright Elsevier, 2005).

**Figure 12 nanomaterials-11-00019-f012:**
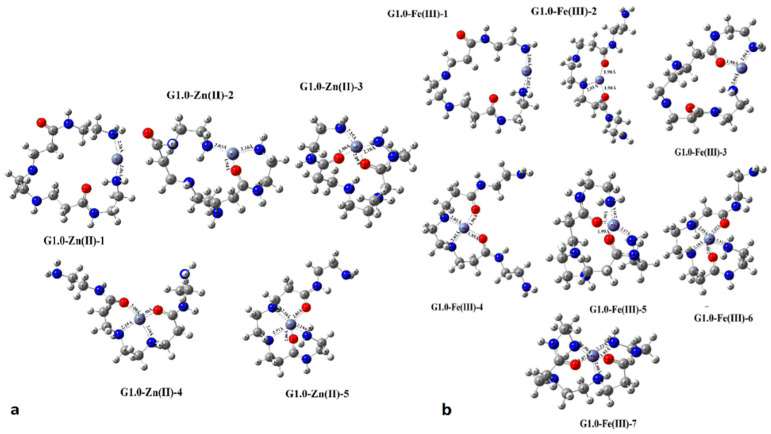
Optimized coordination structures of PAMAM G1.0 with (**a**) Zn(II) (Reproduced with permission from [98]; Copyright Elsevier, 2020) and (**b**) Fe(III) (Reproduced with permission from [103]; Copyright Elsevier, 2018).

**Figure 13 nanomaterials-11-00019-f013:**
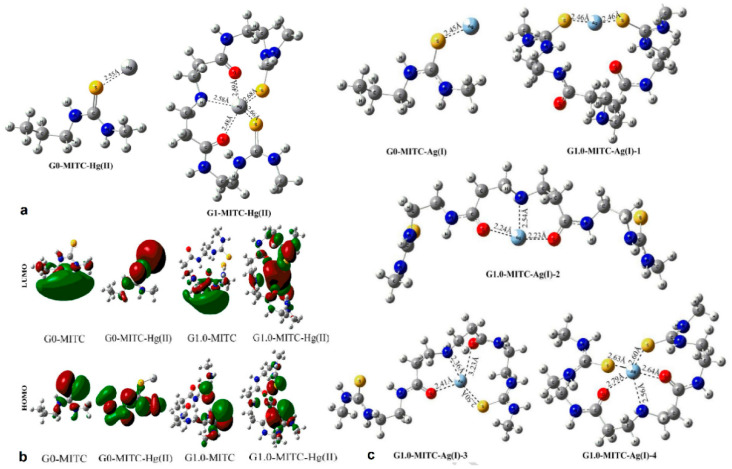
Optimized geometries (**a**) and contour plots (**b**) of the highest occupied molecular orbitals HOMOs (red) and lowest unoccupied molecular orbitals LUMOs (green), of Hg^2+^ complexes with methylisothiocyanated (MITC) G0-PAMAM and G1.0-PAMAM. (Reproduced with permission from [110]; Copyright American Chemical Society, 2016); (**c**) Optimized geometries of the respective silver complexes (Reproduced with permission from [111]; Copyright Elsevier, 2018).

**Figure 14 nanomaterials-11-00019-f014:**
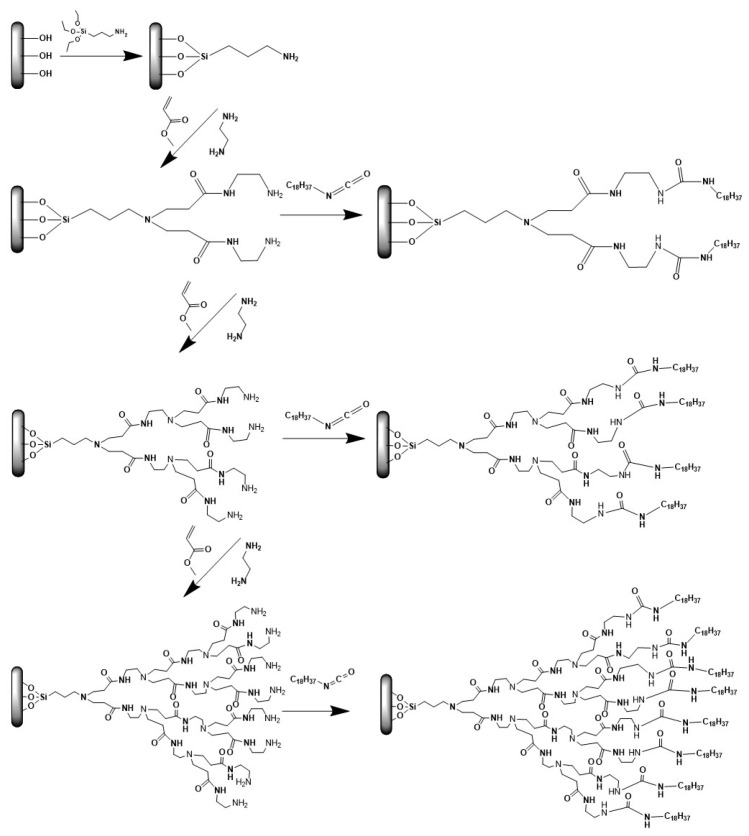
Formation of PAMAM dendrons on silica surface and generation of amphiphilic dendron surface-modified silica after functionalization of amino-end groups with octadecylisocyanate (Reproduced with permission from [97]; Copyright American Chemical Society, 2008).

**Figure 15 nanomaterials-11-00019-f015:**
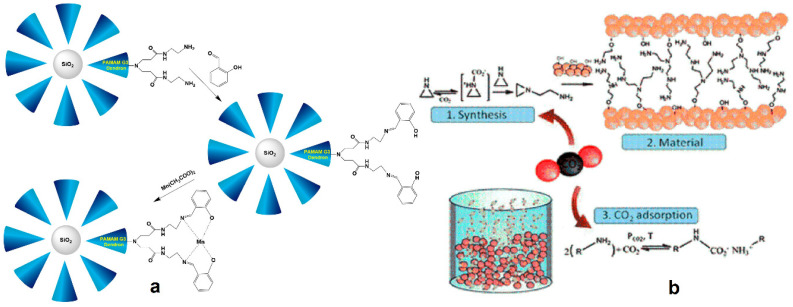
(**a**) Procedure for assembling Silica-PAMAM Dendron-Manganese Composite Catalysts. (**b**) Schematic representation of PEI dendron growth onto silica using CO_2_, and the mechanism of posterior CO_2_ adsorption (intended application) (Reproduced with permission from [127]; Copyright Royal Society of Chemistry, 2013).

**Figure 16 nanomaterials-11-00019-f016:**
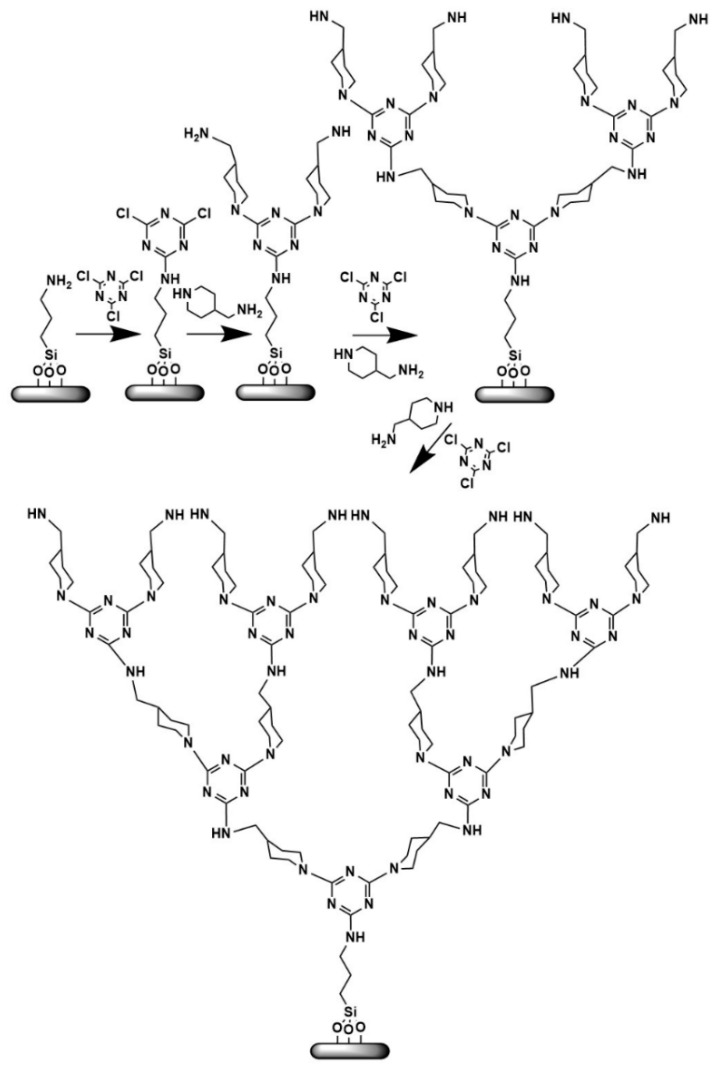
Evolution stages of melamine-based dendrons (Reproduced with permission from [128]; Copyright WILEY-VCH Verlag GmbH & Co., 2004).

**Figure 17 nanomaterials-11-00019-f017:**
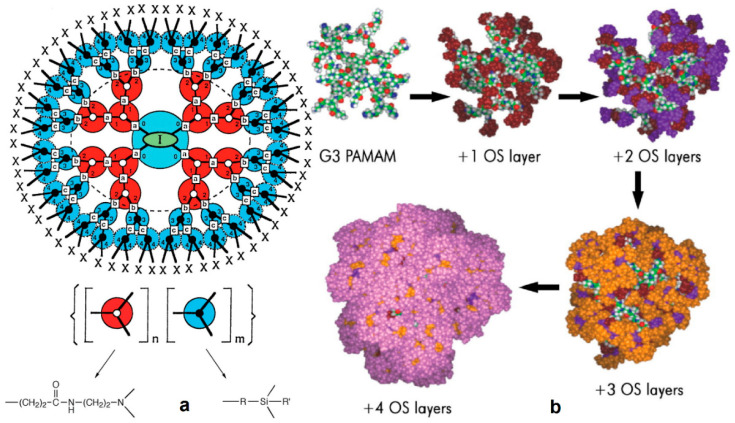
(**a**) Generalized representation of PAMAMOS dendrimer structure I: core; X: end-groups; 1, 2, 3, 4, denote generations; letters: a, PAMAM-PAMAM bonds b, PAMAMOS bonds c, OS-OS bonds. (**b**) Molecular models of Generation 3 PAMAM interior (green–red–white–blue) with increasing number of organosilicon exterior layers. First layer brown; 2nd layer purple; 3rd layer yellow; 4th layer magenta (Reproduced with permission from [133]; Copyright WILEY-VCH Verlag GmbH & Co., 2006).

**Figure 18 nanomaterials-11-00019-f018:**
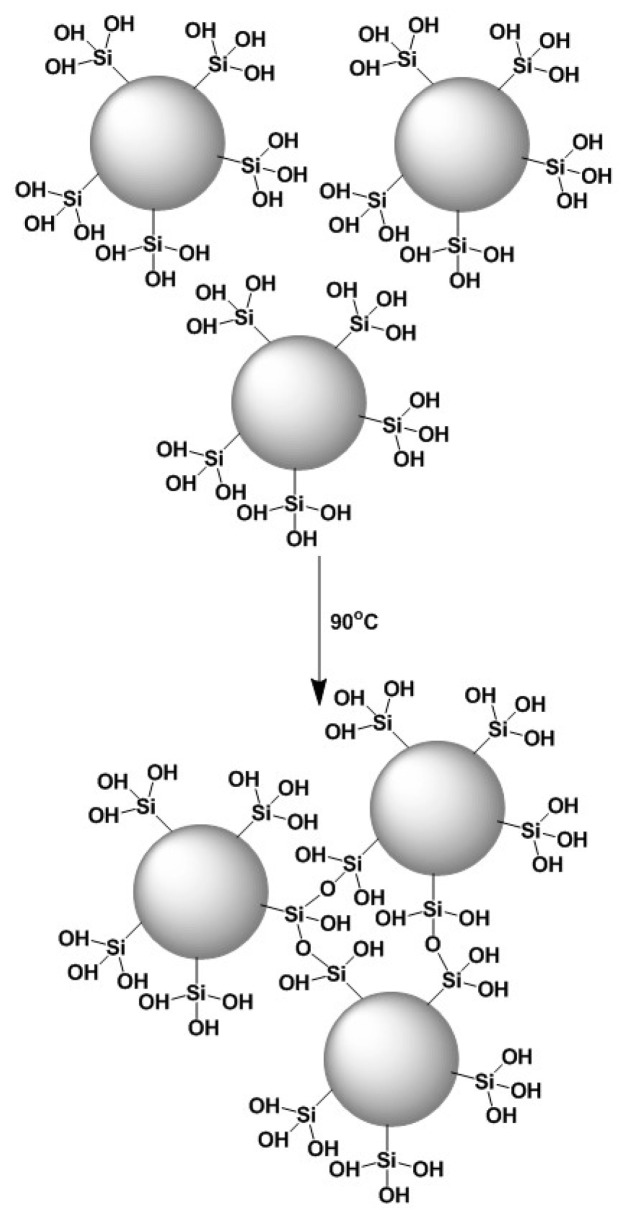
Formation of honeycomb-like PAMAMOS networks from silyl dendrimer precursors (Reproduced with permission from [133,134]; Copyright WILEY-VCH Verlag GmbH & Co., 2006; American Chemical Society, 2002).

**Figure 19 nanomaterials-11-00019-f019:**
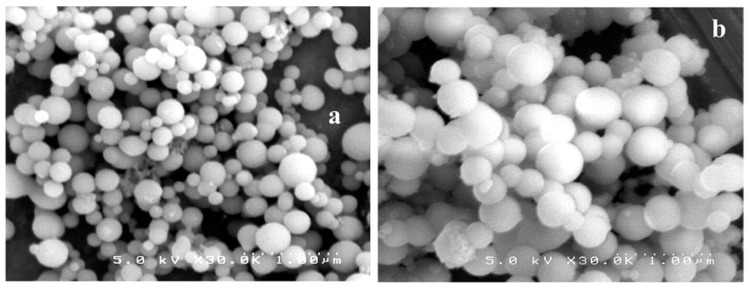
SEM micrographs of silica nanospheres, biomimetically templated by Amine-Terminated Dendrimers (**a**) G4 PPI and (**b**) G4 PAMAM (Reproduced with permission from [142]; Copyright American Chemical Society, 2004).

**Figure 20 nanomaterials-11-00019-f020:**
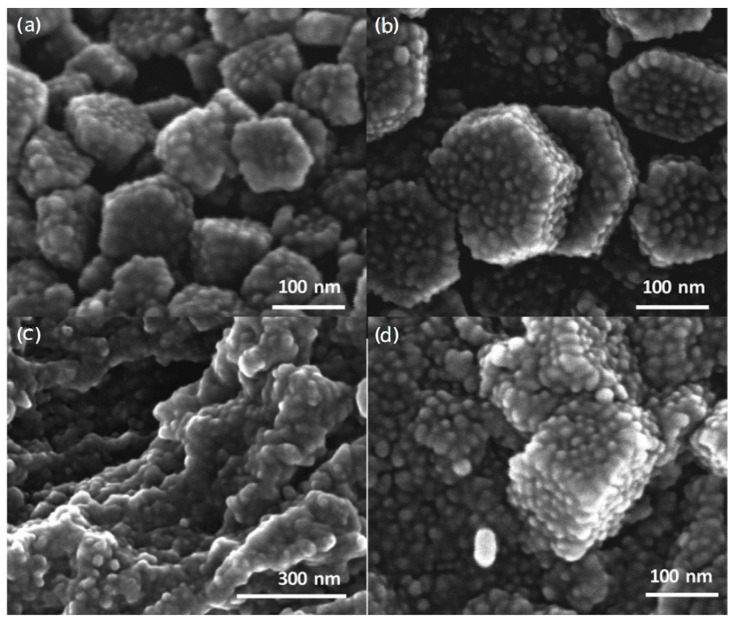
SEM micrographs of alumina nanospheres templated by hyperbranched PEI for different quantities of nano-dispersed boehmite ((**a**) 0.02, (**b**) 0.04, (**c**) 0.06, and (**d**) 0.08 g/mL) (Reproduced with permission from [146]; Copyright The American Ceramic Society, 2018).

**Figure 21 nanomaterials-11-00019-f021:**
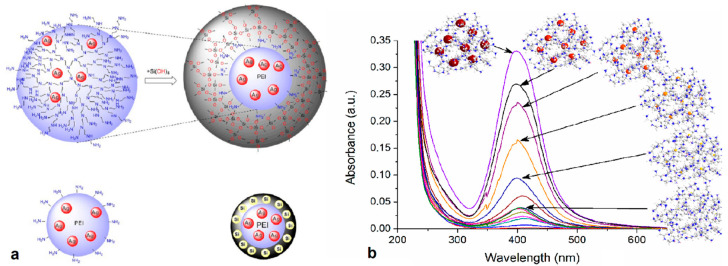
(**a**) Schematic representation of hybrid silica-PEI-Ag nanoparticles; (**b**) UV-Vis spectra of silica-PEI-Ag nanoparticle suspension in different time periods (Reproduced with permission from [150]; Copyright Elsevier, 2018).

**Figure 22 nanomaterials-11-00019-f022:**
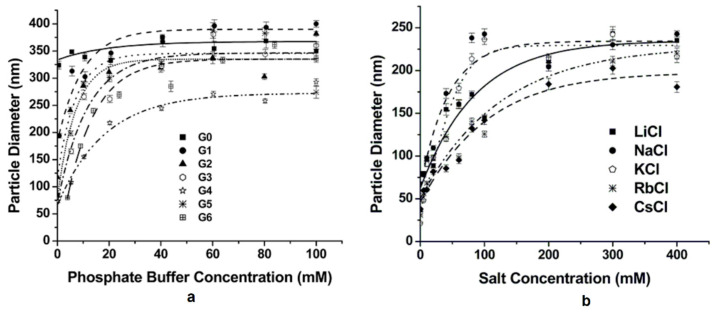
(**a**) Particle size of nanospheres produced by different PAMAM generations as a function of phosphate buffer concentration (**b**) as a function of salt concentration for different alkali metal chlorides (Reproduced with permission from [143]; Copyright American Chemical Society, 2005).

**Figure 23 nanomaterials-11-00019-f023:**
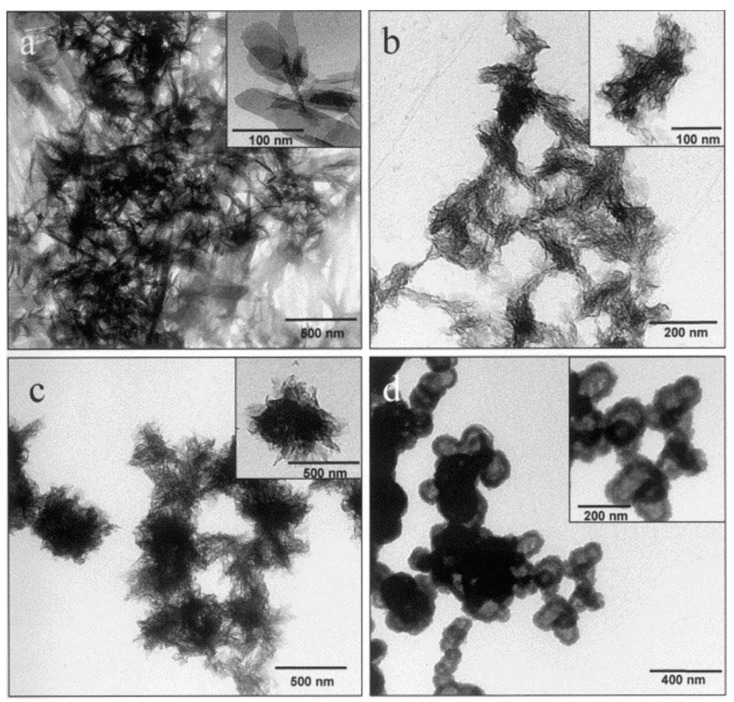
TEM image of hydroxyapatite prepared by self-precipitation from SBF. (**a**) Direct precipitation from SBF at pH 9.0 in absence of dendrimer. (**b**) Self-precipitation in the presence of 4.5 generation PAMAM dendrimer from SBF at pH 7.4. (**c**) Self-precipitation in the presence of 3.5 generation PAMAM dendrimer from SBF at pH 8.5. (**d**) Self-precipitation in the presence of 3.5 generation PAMAM dendrimer from SBF at pH 8.0. Insets in all figures show typical hydroxyapatite particle shape and morphology under magnification (Reproduced with permission from [146]; Copyright The American Ceramic Society, 2018).

**Figure 24 nanomaterials-11-00019-f024:**
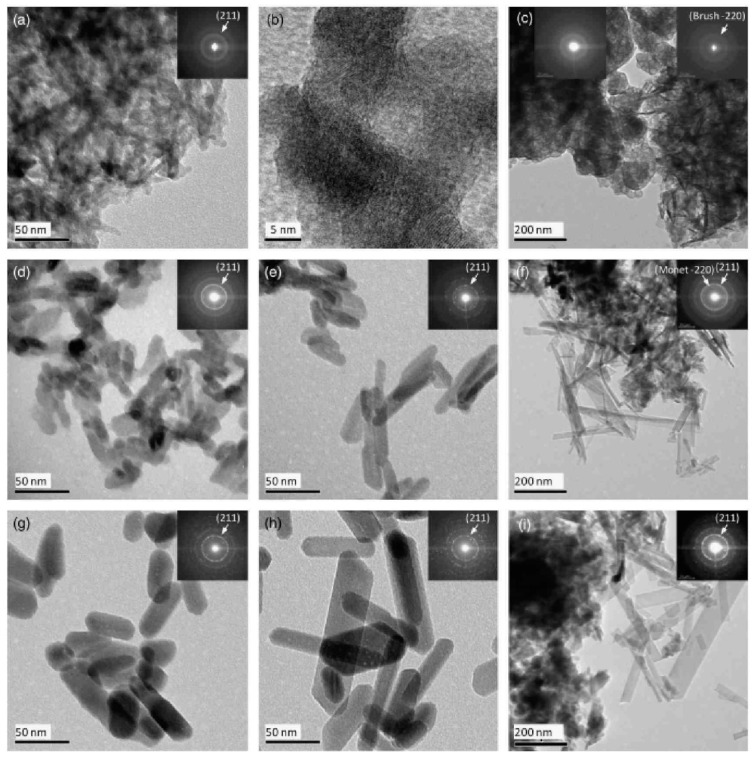
TEM images and respective selected area electron diffraction patterns of nanoparticles obtained under different conditions. Upper row for samples prepared at room temperature: (**a**) HAP dendrimer:calcium 1:1, (**b**) high resolution image, and (**c**) control (without DAB); middle row for samples hydrothermally treated at 80 °C: (**d**) HAP dendrimer:calcium 2:1, (**e**) HAP dendrimer:calcium 1:2, and (**f**) control; lower row for samples hydrothermally treated at 130 °C: (**g**) HAP dendrimer:calcium 2:1, (**h**) HAP dendrimer:calcium 1:2, and (**i**) control (Reproduced with permission from [160]; Copyright The American Ceramic Society, 2011).

**Figure 25 nanomaterials-11-00019-f025:**
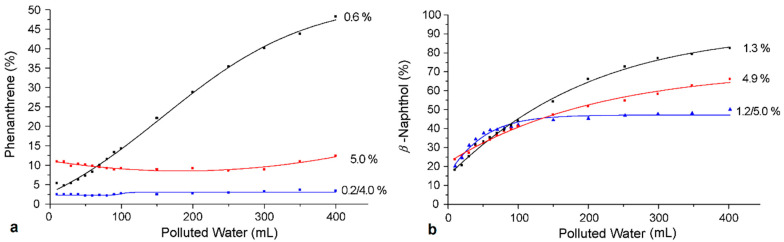
Percentage of phenanthrene (**a**) and β-naphthol (**b**) remaining in water in continuous filtration through filters impregnated by different percentages of octyl-substituted hyperbranched PEI employing a single or a two-step immersion stage (Reproduced with permission from [61]; Copyright Elsevier, 2008).

**Figure 26 nanomaterials-11-00019-f026:**
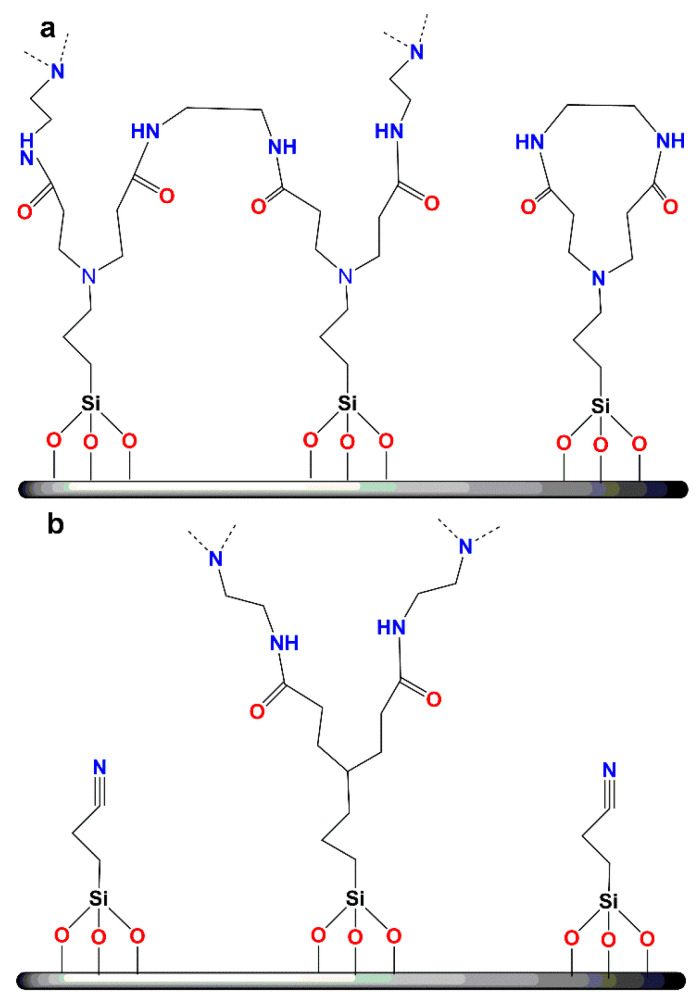
(**a**) Common structural defects occurring during the dendron propagation reaction; (**b**) introduction of triethoxysilyl propylonitrile as a spacer for the elimination of cross-linking structural defects (Reproduced with permission from [96]; Copyright American Chemical Society, 2004).

**Table 1 nanomaterials-11-00019-t001:** General applications of ceramics/dendritic polymer composites.

Dendritic Polymer-Ceramic	Application	Reference
PPI-SiO_2_	Electrochemical (Redox-Active Materials)	[65]
PAPAM-PDMS	High-Resolution Contact Printing	[68]
Benzyl-Terminated Dendrons-SiO_2_	High-Resolution Capillary Gas Chromatography	[82]
Benzyl-Terminated Dendrons-SiO_2_	Capillary microextraction	[83]
L-glutamic acid-Chiral-Dendrimers-SiO_2_	HPLC	[90]
PAMAM-SiO_2_	Photoluminescence Sensors	[91]
PAMAM-TiO_2_	TiO_2_ Nanoparticle Stabilization	[92]
PAMAM-SiO_2_	Size Exclusion Chromatography	[108]
Melamine Dendrons-Al_2_O_3_	Polarity-Based Separation	[130]
PAMAMOS	Electronics, Photonics, Magnetics, Sensors, Coatings	[133,134,135]

**Table 2 nanomaterials-11-00019-t002:** Applications of ceramics/dendritic polymer composites in catalysis.

Dendritic Polymer-Ceramic	Application	Reference
PEI-SiO_2-_TiO_2_	Nitro-compounds Reduction	[29]
CBS-SiO_2_	Ethylene Polymerization	[79]
Pyridylphenylene dendrons-SiO_2_	Suzuki cross-coupling	[86]
Chiral-PAMAM-SiO_2_	Enantiomeric Reactions	[87]
DAB-SiO_2_	Selective Hydrogenation	[88]
PAMAM-SiO_2_	Alcohol Oxidation	[113]
PAMAM-SiO_2_	Olefin Hydroformylation	[114,115,116]
PAMAM-SiO_2_	Carbonylation	[117,118,119]
PAMAM-SiO_2_	Hydroesterification	[120]
PAMAM-SiO_2_	Heck reaction	[121]
PAMAM-SiO_2_	Olefin Epoxidation	[122]
PAMAM-SiO_2_	Enzymolysis	[123]
PAMAM-SiO_2_	Benzyl Alcohol, Methylene Blue Oxidation, Nitrophenol Reduction	[149]
DAB-SiO_2_	Phenols Hydrogenation	[171]
PEI-SiO_2_	CO, NO, CH_4_, C_3_H_6_, C_3_H_8_ Oxidation	[172]
PEI-SiO_2_-CeO_2_	Nitrophenol Reduction	[173]

**Table 3 nanomaterials-11-00019-t003:** Adsorption properties of ceramics/dendritic polymer composites.

Dendritic Polymer-Ceramic	Application	Reference
DAB, PEI, PG-Al_2_O_3_, TiO_2_ SiC	PAH, BTX, THM, MTBE, Pesticides	[58,59]
PEI-Al_2_O_3_	PAH	[61]
PAMAM-SiO_2_	Fe_4_[Fe(CN)_6_]_3_, Co_2_[Fe(CN)_6_]	[62]
PAMAM-TiO_2_	Cu(II), Ni(II), Cr(III)	[67]
PAMAM-SiO_2_	Hg(II)	[80,109,110,112]
PPH-SiO_2_	CO_2_	[84]
PEI-SiO_2_	Cu(II), Zn(II), Cd(II)	[85]
PAMAM-SiO_2_	Cu(II)	[94]
Melamine Dendrons-SiO_2_	Atrazine	[95]
PAMAM-Dendrons-SiO_2_	Organic Dyes, Amphiphilic Surfactants	[97]
PAMAM-SiO_2_	Zn(II), Co(II)	[98]
PAMAM-SiO_2_	Mn(II)	[99]
PAMAM-SiO_2_	Ni(II)	[100]
PAMAM-SiO_2_	Co(II)	[101]
PAMAM-SiO_2_	Cd(II)	[102,103]
PAMAM-SiO_2_	Fe(III)	[103,104]
PAMAM-SiO_2_	Pb(II)	[105]
PAMAM-SiO_2_	U(IV)	[106]
PAMAM-SiO_2_	Ag(I)	[111,112]
PEI-SiO_2_	CO_2_	[125,126]
TREN-SiO_2_	CO_2_	[129]
PEI-SiO_2_	PAH, Pb(II), Hg(II), Cd(II), Cr(VI)	[148]

**Table 4 nanomaterials-11-00019-t004:** Ceramics/dendritic polymer composites applications in nanomedicine.

Dendritic Polymer-Ceramic	Application	Reference
PEI-HA	Hydroxy apatite formation	[30]
PAMAM-SiO_2_	Gene therapy	[63]
PAMAM, CBS-SiO_2_	Antibacterials	[77]
CBS-SiO_2_	Oligonucleotide Delivery Carriers	[78]
PAMAM-SiO_2_	Fluorescence Imaging	[81]
PPH-SiO_2_	Antibacterials	[89]
PAMAM-SiO_2_	Anticancer Photothermal Therapy	[124]
PEI-SiO_2_	Orthopedic Implants	[138]
PEI-SiO_2_	Antibacterials	[150]
PAMAM-HA	Hemoglobin Aquasomes	[156]
PAMAM-SiO_2_	Gene transfection	[175]

**Table 5 nanomaterials-11-00019-t005:** Applications of ceramics/dendritic polymer composites in drug delivery.

Dendritic Polymer-Ceramic	Application	Reference
PAMAM-SiO_2_	Doxorubicin, Curcumin	[64]
DAB-SiO_2_	Ibuprofen	[74]
DAB-SiO_2_	Levofloxacin	[75]
DAB-SiO_2_	Redox-responsive release	[76]
PAMAM-SiO_2_	Curcumin	[81]
PAMAM-SiO_2_	Doxorubicin and Bcl-2 targeted siRNA	[124]

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
