# Peer review of "Dendritic Polymers as Promising Additives for the Manufacturing of Hybrid Organoceramic Nanocomposites with Ameliorated Properties Suitable for an Extensive Diversity of Applications"

_nanomaterials, 2020, doi:10.3390/nano11010019_

Round 1
Reviewer 1 Report
A review that covers a very broad range of topics. This can both be a strength and a weakness, because a typical reader will most likely be interested in a specific part of the review. I would recommend a more focused review, which for example either focused on the materials side (bones, teeth, ...) or went into depth catalysis (efficiency, stability, dendritic effects? and so on.
Reviewer 2 Report
According to the title presented review summarizes published data concerning the construction of novel hybrid nanocompounds obtained by combination of suitably functionalized dendritic polymers and ceramic compounds, providing the powerful toos for established and prospected variety of applications. Authors collected a wealth of information from many publications concerning wide array of the various aspects of that problem, and included many reprinted schemes and pictures. This makes the review interesting for researchers searching for that kind of data. Manuscript should be published after suggested improvements and supplementations.
General comments:
Authors are giving much attention to the methods of synthesis and variety of applications of different class of described compounds. However it may help readers if certain details were presented as Tables with respective references (e.g. summarization of described types of hybride composites and methods used for evaluation of their properties as well as introduced and perspective applications). Also medical aspects of presented compounds should be described in separate paragraph. This should include the function of dendrimers as carriers for targeting molecules (small molecules, vitamins, antibodies), very important issue including imaging, diagnostics and therapeutic application of perspective hybride constructs since the nanomedicine is considered as the future of medicine (e.g. Yetisgin et al. Molecules 2020,25,2193; Shahbazi-Gahrouei et al. Res Med Sci. 2019;24:38; Le et al. Pharmaceutics. 2019;11(11):591; Shukla et al.. Polymeric Nanoparticles as a Promising Tool for Anti-cancer Therapeutics . Academic Press .2019. 233-255; ISBN 9780128169636, Shi et al. Analyst. 2009;134(7):1373-1379; Fan et al. Small Methods 2017, 1 (12) , 1700224.; Devnarain et al. . Wiley Interdiscip Rev Nanomed Nanobiotechnol. 2020:e1664. doi: 10.1002/wnan.1664; Majumder, Minko. Expert Opin Drug Deliv. 2020. 8:1-23; Saravanakumar et al. Current Pharmaceutical Design.2019. 25: 2609; Li et al. Wiley Interdisciplinary reviews. Nanomedicine and Nanobiotechnology. 2020:e1670. DOI: 10.1002/wnan.1670).
Majority of citations come from before 2018 year, whereas many new appeared. At least two important recent references should be included : Li et al. Wiley Interdisciplinary Reviews Nanomedicine and Nanobiotechnology. 2020. DOI: 10.1002/wnan.1658; Castillo,Vallet-Regi. Int. J. Mol. Sci. 2019, 20, 929.
Other comments:
Original Figures descriptions are not enough informative (Figs.6 a and b, 10, 14, 16, 18 and 21). Certainly that schemes and pictures were drown on the basis of data from publications by the other authors , so respective references should be included. Certain pictures are too small for recognition.
- (Figure 6 up) ?
- Addition of n-octadecylisocyanate, leads to the introduction of long aliphatic chains to the periphery of the dendrons – reference?
- „Absorption and crosslinking of external amino groups with cellulase provides enzymolysis activity” – should be rather: stabilizes and enhances enzymolysis effeciency
- „Bcl-2-targeted” : requires explanation of the abreviation
485 „hafnia” is is a psychrotrophic bacterium
Reviewer 3 Report
Manuscript describes a class of inorganic/organic hybrid ceramics where organic part involves branched molecules, such as well structurally defined dendrimers and dendrons and also dendrimers with no perfect structure of branches called dendritic polymers.
Authors focused on methods of preparation and on the prospective applications of such nanomaterials in areas of orthopedic therapy, separation technology, medical hydrogels, etc. Manuscript is logically assembled, interesting and supplemented with nice figures. Although as a whole it is well understood, introductory part needs to be improved. It has some imperfection in logic and vocabulary that have to be corrected before publication as listed below:
Line 32: “..nanoscale analogues..”; imprecise, analogs of what?
Line 35:”… designification…”; what you mean?
Line 36: “The exceptionality of these materials originates from their multilayer, core-shell-surface architecture, and depending on their core substance, they can be classified in three key categories; ceramic nanocomposites, metal nanocomposites and polymeric nanocomposites [1–3].” Make 2 sentences,
Further: Please make strict separation into two classes: dendrimers and dendritic polymers. First group constitutes monodispersed nanomolecules synthesized by using simple organic reactions, well purified and well characterized structures. Second group constitutes polymers with branched architecture that even if they have similar size are a collection of molecules with different chemical structures.
Line 88: “… crystallization degrees characterized by high bioactivity, biodegradability and resorbability and are ideal for repairing purposes, when a strong and controlled bond with the implant is required…”. What you mean???
Line 106: “They constitute either monodispersed symmetric dendrimers named after the Greek word δÎνδρο for tree because of their structural resemblance or asymmetric species, called hyperbranched polymers, fragments of both categories defined as dendrons or even dendronized forms, which are combinations of conventional polymers having similar patterns. Not clear!
Line 110: Replace “radial polymerization” by “radial functionalization”, since most often conventional chemical methodology is applied for their synthesis.
Authors are using term “dendritic polymers” and they often mean well defined dendrimers. Dendritic polymer is a polymer containing branched structures, most often arranged in a statistical not stoichiometric manner!
Line 153: “.. properties, due to the highly accessible empty cavities inherited by the multi-branched configuration of the dendritic compound.” Multi-branching is not “configuration” in a chemical sense. Please correct.
Line 199: Although Figure 4 was reprinted from ref 61 the metal coordination structure is misleading. Make your own Figure or simply call Fig. 12.
Line 242: ”… Some of the most prominent advantages of these hetero-atom dendrimers over the conventional ones are… “; What do you mean. The most common dendrimers are PAMAM’s, PPI peptide dendrimers, etc. that undoubtfully contain heteroatoms.
Line 275: “…Interference of disulfide bridges between G1-G5 PPI and mesoporous silica nanoparticles generated redox-responsive stimuli triggered release carriers for fluorescent dyes (fluorescein disodium salt and carboxyfluorescein) [70].” Please be more precise on what is the source of sulfur.
Line 418: “… PAPAM-filled mesoporous silica particles were used as well for generation dependent size exclusion chromatography columns…” PAMAM(?); “as filing of column(?)”.
Line 572: are encapsulated into the growing nanospheres or conjugated???
Line 635: Figure 22. (a) Particle size of nanospheres produced by different PAMAM generations as a function of phosphate buffer (b) as a function of salt concentration for different alkali metal chlorides…” Add “buffer concentration”.
Concluding remarks and perspectives: Please state more precisely which are the main obstacles in application of such nanocomposites in medicine and technology and also future directions in their preparation and characterization.
Reviewer 4 Report
The Authors reviewed the subject of solid state materials, specifically the surface modification of inorganic oxides like silica, alumina, titania and others with organic dendritic molecules. The subject seems to be interesting considering a potential applications of hybrids materials for separation techniques due to tunable surface of hybrid. Because I am inorganic chemist in origin, but working on application of dendrimers in drug delivery systems the review was especially valuable for me to read and comment it from another point of view than Authors, who perform their research on inorganic/organic hybrid adsorption.
Considering the Author’s concepts, I would expect to comment the following issues:
1.What advantageous properties have silica modified with organic dendrimers in comparison with dendrimers themselves ?
It is not for catalysis, because the catalytic surfaces or cavities of dendrimers are available in homogeneous conditions. Compare for example:
Homogeneous Pd or Rh in dendrimer for catalysis: Pittelkow et al. Chiral dendrimer encapsulated Pd nad Rh nanoparticles. Chem Commun. (2208) 2358-2360. Shmitzer et al: Reactivity at the interface of chiral amphiphilic dendrimers. High asymmetric reduction by NaBH4 of various prochiral ketones. J Am Chem Soc (2001) 123, 5956-5961,
versus heterogeneous catalysis: [81], [82] and [107-113, 143, 165-167]
My hypothesis is: Immobilization of organic dendrimers on inorganic support converts the homogeneous catalysts into less efficient heterogeneous analogues.
Please comment it in the revised version.
2.The concept of using silica or any other inorganic nanoparticles as drug delivery systems is exceptionally wrong idea in my opinion, consulted many times with medical colleagues. The simple fact is that inorganic nanoparticles like silica or titania cannot be metabolized (they are not biodegradable). Once they are administered in blood circulation system and they reach the targeted cells, they will be deposited in the tissue. The dramatic example is asbestos, which was used as thermal insulator in buildings. This inorganic stuff forms nanosized needles, which after inhalation reach the lung cells, cause their inflammatory response. The cells are eradicated by immune system, while magnesium silicate needles does not move anywhere and invades other cells. Finally they are deposited in lung tissue as fibrotic, physiologically dead tissue. The X-ray pictures of such lungs are commonly known for med doctors. The same problem is due to microparticles of smog (PM 2.5), which cannot be removed from our bodies once they are deposited in lung tissue; the post-mortal examination of lung and other tissues evidences it in obvious way.
Once the silica or other inorganic nanoparticles get into tissue, they cannot be removed by kidney glomerular filtration (when sized > 10 nm). So there is only one way for nanoparticles to be secreted and it is long way to liver tissue via blood circulation system, and then followed by biliary excretion.
Similar problem seems to be the application of organic dendrimers, which are flexible molecules. When they are sized up to ca 7 nm diameter, they can pass the glomerular barrier, otherwise they circulate in blood for quite long time. Therefore PAMAM dendrimers of generation 5 (7 nm diameter) are probably the best drug-delivery system, because they are removed slowly by kidney filtration. Anyway, the biodegradation of organic dendrimers have to be evidenced for future use of dendrimer-based DDSs.
Surprisingly to me the attempts to use the inorganic materials as DDS are still on agenda, probably because they show promising drug release profiles. See for instance paper cited as [118].
So, the question 2 is:
What is the advantage of using silica support for organic dendrimer-drug conjugates or encapsulates as DDS?
- The size of silica nanoparticles are not under control if obtained by sol-gel method. The proposed PAMAMOS [127] seems to progress in that matter, although the protocols used in [127-132] are still based on spontaneous 3-D organization of final materials. Systematic studies on otherwise spontaneous and unpredictable self-assembling are shown in [144] for ternary systems: silica:PEI:AgNP. Similarly for binary system: PAMAM:silica, and variable aqueous conditions.
The general problem is analytical methods: all these composites are not soluble; so the only method to identify the structure is solid state imaging (either be SEM/TEM or AFM) and suspension characterization by zeta potential. This is clearly demonstrated by papers dedicated to apatite formation in presence or absence of dendrimers cited as [148-155].
4.Biomimetic abilities of diatoms and sponges; we have lost these abilities evolutionarily and finally we are not able to organize Si-O-Si framework. So, the silica is chemically inert for humans, neither it is bioavailable from environment. Silica nanoparticles seems the problem for human body; once they are in tissue, they have to be removed.
Remark 4. Please comment above
- In [162] something is missing (Literature)
- Specific remarks:
The narration is very enthusiastic at the beginning (Introduction), with a lot of adjectives (for instance lines: : line 65 – brilliant, line 75 – excipient means, line 81 – powerful type of materials, line 240 – they offer ideal active sites…, line 489 – fascinating subcategory ) and long, comma-separated sentences (for instance lines: 66-73, 101-105, lines 106-109, lines 110-117, 624-626, 705-707, 723-725). Empty sentences could be removed in verified version; see line 765-766: “Thee use of dendritic polymers in now widely accepted.” Generally speaking, the narration is too much talkative.
Authors built the sections starting from statements-judgements, and then they cite the published facts and that is exactly the opposite narration used commonly.
The paper should be considerably reorganized by replacement long sentences with short ones and less adjectives. Some odd phrases like (line 150: spawn hybrids, line 176: pollutant inclusion constant, line 197: validating its high thermal stability, line 207: the pH the.., 208: affect charge of the molecules, line235: etching employing strong acid.., line 266: induction of hydroxyl groups…, line 294: malleable external surface…, whole sentence lines 294-296, line 305: keenly exploited…, line 326-327, line 345: auspicious disciple, line 381: transporting the mixture in water, line 384: their photocatalytic activity towards 2,4-dichlorophenoxyacetic acid; rather some reaction was catalyzed; line 392: are even commercially available…; groups are not commercially available, and besides why does it matter ?; line 424: supplementary stage.., line 468: primordial amino groups, line 476: The yielding modified membrane was further…, line 652: Besides of molding imaginative ceramic nanostructures…, line 688: with broad particle size; “distribution” might be better ) should also be verified. Probably ternary instead of triune (line 788) is more chemical.
Round 2
Reviewer 1 Report
The manuscript has been revised according to the comments and is publishable now.
Author Response
Thank you very much for your time
Reviewer 2 Report
All corrections and supplementations were done properly. However minor language improvements are still required e.g.should be:114: That prompted nanotechnologists, 553: incubated with the solutions, Table 5:Doxorubicin and Bcl-2-targeted siRNA
Author Response
Corrections done. Thank you very much for your time.
Reviewer 4 Report
Thank you, you have done good job. Especially I like the additional Tables, which meka the paper well organized.
Only one sentence could be still shorter; it is the added fragment of paper: lines 953-956 could be divided into at least 2 sentences.
Generally I think it is valueable review paper and I have learn a lot from it.
Thank you
Author Response
The added fragment has been divided into two more sentences. Than you very much for your time.